# MotionTTT: 2D Test-Time-Training Motion Estimation for 3D Motion Corrected MRI

**Tobit Klug**[1,*]    **Kun Wang**[1,*]    **Stefan Ruschke**[2]    **Reinhard Heckel**[1,†]

[1]School of Computation, Information and Technology, Technical University of Munich
[2]Department of Diagnostic and Interventional Radiology, Klinikum rechts der Isar,
School of Medicine and Health, Technical University of Munich

## Abstract

A major challenge of the long measurement times in magnetic resonance imaging (MRI), an important medical imaging technology, is that patients may move during data acquisition. This leads to severe motion artifacts in the reconstructed images and volumes. In this paper, we propose MotionTTT a deep learning-based test-time-training (TTT) method for accurate motion estimation. The key idea is that a neural network trained for motion-free reconstruction has a small loss if there is no motion, thus optimizing over motion parameters passed through the reconstruction network enables accurate estimation of motion. The estimated motion parameters enable to correct for the motion and to reconstruct accurate motion-corrected images. Our method uses 2D reconstruction networks to estimate rigid motion in 3D, and constitutes the first deep learning based method for 3D rigid motion estimation towards 3D-motion-corrected MRI. We show that our method can provably reconstruct motion parameters for a simple signal and neural network model. We demonstrate the effectiveness of our method for both retrospectively simulated motion and prospectively collected real motion-corrupted data. Code is available at `https://github.com/MLI-lab/MRI_MotionTTT`.

## 1 Introduction

Magnetic resonance imaging (MRI) is one of the most important medical imaging technologies due to its non-invasiveness and ability to foster diagnosis of a wide range of diseases. However, its inherently long scan times make MRI susceptible to motion artifacts caused by patient movement during the scan. Repeating scans corrupted by motion artifacts causes additional costs and reduces patient through-put, and unnoticed artifacts can lead to misdiagnosis [2, 36].

We consider the problem of imaging under motion and propose an algorithmic solution to correct for the motion based only on the measurements acquired during the scan, without requiring additional hardware or changing the measurement process, or interrupting the clinical workflow.

A traditional approach to algorithmic motion reconstruction is to jointly estimate the motion parameters and motion-corrected reconstruction [7, 6, 15], but those methods are slow and can be inaccurate in particular for severe motion.

Deep learning based approaches have been proposed to accelerate reconstruction and potentially allow to account for more severe motion. However, most existing data driven methods correct for in-plane motion within 2D MRI (see the review Spieker et al. [38]), as 3D data for training 3D models is only scarcely available and computationally expensive to handle [18]. In practice, however, motion occurs in 3D and not in-plane. Moreover, the duration of a scan in 3D is significantly longer than in 2D making motion more likely and thus motion reconstruction more important. Finally, data-driven

---

*Shared first authors in alphabetic order. †Corresponding author: reinhard.heckel@tum.de

38th Conference on Neural Information Processing Systems (NeurIPS 2024).

approaches so far often rely on simulated motion artifacts and hence are specific to the type of motion they have been trained on [14, 16, 34].

In this work, we propose a novel approach for rigid motion estimation and reconstruction in 3D MRI that is based on first estimating the motion parameters that describe the map from the motion-free image to the motion-corrupted measurement and second reconstructing the image or volume with the estimated motion parameters. Estimating the motion parameters is the critical step, once we know the motion parameters, reconstruction essentially amounts to reconstructing from a motion-free measurement, and a variety of approaches work well for that.

For motion estimation, we utilize a neural network trained to reconstruct motion-free undersampled 2D MRI images. The network only requires 2D motion-free data for training, and does not require 3D or motion-corrupted data which is difficult to come by. The neural network for reconstruction depends on the forward model which in turn depends on the motion parameters. We construct a data-consistency loss and optimize over the motion parameters at test-time. The data consistency loss is small only for the correct motion parameters, as the model was trained for motion-free reconstruction.

In each iteration the model reconstructs the 3D data slice-wise along a random axis. Since motion artifacts occur globally in the image domain it is sufficient to compute gradients only for a small random subset of slices keeping the computational and memory cost manageable. The estimated motion parameters are then used to reconstruct a clean volume. To summarize, our contributions are:

- We propose MotionTTT, the first deep learning-based method for 3D rigid motion estimation for 3D motion-corrected MRI. MotionTTT exploits the prior knowledge of a pre-trained neural network for motion-free 2D MRI reconstruction.

- We theoretically justify our method by proving for a simple theoretical signal and neural network model that the loss function has a global minimum at the correct motion parameters.

- We use retrospectively simulated motion to demonstrate the ability of our method to accurately estimate motion over a wide range of motion severities. Combined with a L1-minimization reconstruction module we achieve effective 3D imaging under motion and outperform a classical alternating optimization baseline [7] in terms of estimation speed and estimation performance under severe motion and a deep learning based end-to-end motion correction baseline Al-Masni et al. [1] in terms of reconstructed image quality.

- We demonstrate the potential of our method on prospectively acquired real motion-corrupted data achieving significant improvements in terms of visual image quality.

## 2 Related work

Approaches for retrospective rigid-motion correction for MRI can be categorized into supervised deep learning-based approaches, model-based optimization approaches, and combinations thereof.

Supervised deep-learning based approaches learn a mapping from undersampled measurements corrupted with simulated motion to the motion-free reference image and have been proposed for 2D [21, 24, 42, 35] and 3D MRI [18, 10, 1]. However, as demonstrated in our work the images produced by those end-to-end approaches are often blurry and the methods pertain to the type of motion simulated during training [23, 14, 16].

Alternating optimization [7] is a classical model-based approach for joint motion estimation and correction in 3D MRI, where every iteration alternates between optimizing over the motion parameters while fixing the estimated reconstruction and vice versa. However, jointly optimizing over both unknowns without any prior information is a highly complex optimization problem resulting in long run times and errors in the presence of more severe motion as demonstrated in our work.

The speed and robustness of alternating optimization can be improved through augmenting either the reconstruction step or the motion estimation step with deep learning [14, 16]. However, both methods have so far only been proposed for 2D in-plane motion correction and rely on synthetic motion simulation during training, which makes them specific to the type of simulated motion. In practice, motion always occurs in 3D.

For motion estimation, Singh et al. [34] pre-trains a neural network to predict a motion-free image from the motion-corrupted measurements conditioned on the true motion parameter. At inference

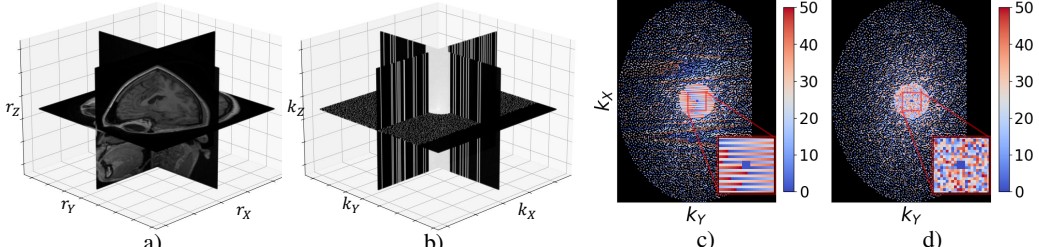

Figure 1: Panel a): magnitude of a 3D volume; panel b): the corresponding 3D k-space data. Panels c)/d) show examples of undersampling masks used for the simulated and real data. The color coding illustrates an interleaved c) and a random d) sampling trajectory indicating which lines along the readout dimension $k_z$ are sampled within the same out of 50 shots.

the data consistency loss is optimized only over the motion parameters. However, the motion estimation relies on learning the characteristic of motion-corruptions. Hence, the model operates in the measurement space in which the corruptions occur, which in multi-coil MRI is of much higher dimension than the corresponding image space, thus making an extension of this approach to 3D very challenging. Our method relies on learning the characteristics of motion-free data in the image space and thus is efficent for 3D imaging as we show.

Levac et al. [22] proposes a method based on diffusion models trained on generating motion-free images for 2D motion reconstruction, that also does not rely on motion simulation during training. At inference, sampling based reconstruction [17, 5] with joint motion trajectory estimation is performed.

While our approach as well as the aforementioned works correct motion artifacts solely based on the acquired MRI measurements, another line of research corrects for motion prospectively or retrospectively based on additional data collected during the scan via, e.g., external detectors [30, 27] or navigator sequences [40, 43, 11]. However, those methods usually require interference with the standard clinical measurement process and are often tailored to a specific measurement sequence or setup limiting their broad applicability in practice.

## 3 Problem statement: 3D MRI imaging under motion

A 3D multi-coil accelerated MRI measurement $\mathbf{y} \in \mathbb{C}^{C \times k_x \times k_y \times k_z}$ is obtained by

$$\mathbf{y} = \mathbf{A}\mathbf{x} + \mathbf{z}, \tag{1}$$

where $\mathbf{A} = \mathbf{MFE}$ is the forward model, $\mathbf{x} \in \mathbb{C}^{r_x \times r_y \times r_z}$ the object of interest, and $\mathbf{z}$ measurement noise. The measurement $\mathbf{y}$ consists of $C$-many k-space measurements collected by $C$ coils. The expand operator is defined as $\mathbf{E}\mathbf{x} = [\mathbf{S}_1\mathbf{x}, \ldots, \mathbf{S}_C\mathbf{x}]$ with coil sensitivity maps $\mathbf{S}_j$. The 3D Fourier transform $\mathbf{F}$ and undersampling mask $\mathbf{M}$ are applied coil-wise. We consider 3D Cartesian undersampling, where undersampling takes place in the plane of the two phase encoding dimensions $k_x \times k_y$ and the frequency encoding dimension $k_z$ is fully sampled (see Figure 1).

We focus on rigid motion, where the $i$-th motion state is defined by three translation and three rotation parameters $\mathbf{m}_i = [t_1^i, t_2^i, t_3^i, \phi_1^i, \phi_2^i, \phi_3^i]$. MRI acquisition under motion can be described as

$$\mathbf{y} = \mathbf{A}(\mathcal{T}, \mathbf{m})\mathbf{x} + \mathbf{z}, \tag{2}$$

where the forward model $\mathbf{A}$ is a function of the unknown motion states $\mathbf{m} = [\mathbf{m}_1, \ldots, \mathbf{m}_b]$, where $b$ is the number of motion states, and of the known MRI sequence specific sampling trajectory $\mathcal{T}$ that specifies which part of the k-space data $\mathbf{y}$ is acquired when.

The goal of this work is to reconstruct the volume $\mathbf{x}$ from the undersampled, motion-corrupted measurement $\mathbf{y}$ without knowledge of the motion states. We do so by estimating the motion states $\mathbf{m}$ from the measurement $\mathbf{y}$ and sampling trajectory $\mathcal{T}$ and use the estimated parameters to reconstruct a motion-corrected volume $\hat{\mathbf{x}}$.

**Parameterization of the forward model under motion.** In practice, a measurement is acquired in batches of lines in the k-space along the frequency encoding dimension $k_z$. A batch, referred to as

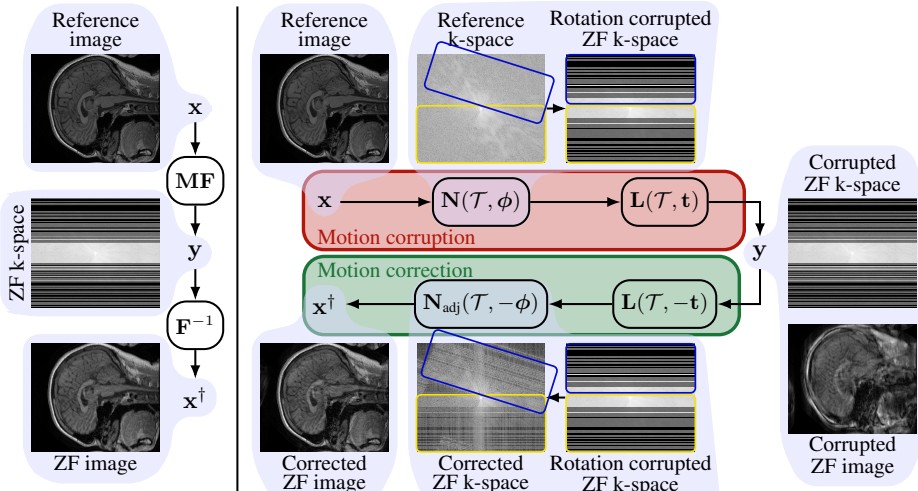

Figure 2: Illustration of the MRI forward models and zero-filled (ZF) reconstructions without (left) and with (right) motion for the 2D single-coil setup. Rotations are implemented with the NUFFT $\mathbf{N}(\mathcal{T}, \phi)$ and adjoint NUFFT $\mathbf{N}_{\text{adj}}(\mathcal{T}, -\phi)$, and translations with linear phase shifts $\mathbf{L}(\mathcal{T}, \mathbf{t})$. During acquisition under rotations areas of the k-space are sampled multiple times while others are not sampled at all, resulting in additional undersampling artifacts in the corrected ZF image compared to the motion-free ZF image.

shot, is acquired within a short time window followed by a pause before the next subsequent shot. It is popular to assume that a subject's position is constant during one shot and motion happens during the pause between shots. This is known as *inter-shot motion* [7, 6, 18, 22, 34, 16]. For inter-shot motion, the number of motions states $b$ is equal to the number of shots and the sampling trajectory $\mathcal{T}$ maps the lines in the k-space acquired during the $i$-th shot to the $i$-th motion state. See Figure 1 (c,d) for examples of sampling trajectories used in practice.

However, in practice, motion can occur anytime, and thus the inter-shot motion introduces an approximation error. Motion during the acquisition of one shot is referred to as *intra-shot motion* [14]. Then, each k-space line acquired during such a shot can have a a distinct motion state. In this work, we investigate the capabilities of our method under both inter- and intra-shot simulated motion.

Within the forward model $\mathbf{A}(\mathcal{T}, \mathbf{m})$, motion corruption can be applied in the image or in the k-space [25], since rotations and translations in the image space translate to rotations and linear phase shifts in the Fourier space. We apply motion corruption in the k-space, because it is computationally more efficient for our setup (see Appendix A). We use the non-uniform FFT (NUFFT) $\mathbf{N}(\mathcal{T}, \phi)$ to sample fully-sampled k-space data at the rotated coordinates for each shot. Translations are applied via linear phase shifts $\mathbf{L}(\mathcal{T}, \mathbf{t})$ to obtain motion-corrupted k-space data.

As a starting point for reconstructing the volume $\mathbf{x}$ from an undersampled measurement $\mathbf{y}$ with zero-filled (ZF) missing entries, it is common to compute a ZF reconstruction $\mathbf{x}^{\dagger} = \mathbf{A}^{\dagger}\mathbf{y}$, where $\mathbf{A}^{\dagger}\mathbf{y} = \sum_{j=1}^{C} \mathbf{S}_j^* \mathbf{F}^{-1} \mathbf{y}_j$. If the measurement $\mathbf{y}$ is motion corrupted, a corrected ZF reconstruction based on motion parameters $\mathbf{m}$ is $\mathbf{x}^{\dagger} = \mathbf{A}^{\dagger}(\mathcal{T}, -\mathbf{m})\mathbf{y}$. First, translations are reverted with a phase shift in the opposite direction $\mathbf{L}(\mathcal{T}, -\mathbf{t})$. Then, rotations are corrected for via the adjoint NUFFT $\mathbf{N}_{\text{adj}}(\mathcal{T}, -\phi)$. See Figure 2 for an illustration of the forward model and ZF reconstruction.

## 4  MotionTTT

The proposed MotionTTT consists of: 1) pre-training a neural network for 2D motion-free image reconstruction, 2) test-time-training to estimate motion parameters of the motion-corrupted 3D k-space data and 3) reconstructing the motion-corrected 3D image based on the estimated motion.

**Step 1: Pre-training.**  Given motion-free data $\{(\mathbf{x}_1, \mathbf{y}_1), \ldots, (\mathbf{x}_N, \mathbf{y}_N)\}$ consisting of pairs of 2D reference images $\mathbf{x} \in \mathbb{C}^{r_x \times r_y}$ and undersampled k-space data $\mathbf{y} \in \mathbb{C}^{C \times k_x \times k_y}$, we train a U-net [32]

$f_{\boldsymbol{\theta}}$ with weights $\boldsymbol{\theta}$ to map a ZF reconstruction $\mathbf{A}^{\dagger}\mathbf{y}_i$ to the image $\mathbf{x}_i$ by minimizing the loss

$$\mathcal{L}_{\text{train}}(\boldsymbol{\theta}) = \sum_{i=1}^{N} \left( \left\| |f_{\boldsymbol{\theta}}(\mathbf{A}^{\dagger}\mathbf{y}_i)| - |\mathbf{x}_i| \right\|_1 / \|\mathbf{x}_i\|_1 + \left\| \mathbf{F}\mathbf{E}f_{\boldsymbol{\theta}}(\mathbf{A}^{\dagger}\mathbf{y}_i) - \mathbf{F}\mathbf{E}\mathbf{x}_i \right\|_1 / \|\mathbf{F}\mathbf{E}\mathbf{x}_i\|_1 \right). \quad (3)$$

We use this combined training loss between magnitude images and k-space data since it leads to better performance for motion-free reconstruction than using one of the individual losses (see Appendix C).

**Step 2: Test-time-training for motion estimation.** Given an undersampled and potentially motion corrupted 3D measurement $\mathbf{y}$ and a sampling trajectory $\mathcal{T}$ we freeze the weights $\hat{\boldsymbol{\theta}}$ of the trained network $f_{\hat{\boldsymbol{\theta}}}$ and estimate the motion parameters $\mathbf{m}$ by minimizing the data consistency loss

$$\mathcal{L}_{\text{TTT}}(\mathbf{m}) = \left\| \mathbf{A}(\mathcal{T}, \mathbf{m}) f_{\hat{\boldsymbol{\theta}}} \left( \mathbf{A}^{\dagger}(\mathcal{T}, -\mathbf{m})\mathbf{y} \right) - \mathbf{y} \right\|_1 / \|\mathbf{y}\|_1, \quad (4)$$

with Adam [19]. The idea behind minimizing this loss is as follows. If applied to motion-corrupted data, at initialization with $\mathbf{m} = \mathbf{0}$ the motion correction $\mathbf{A}^{\dagger}(\mathcal{T}, -\mathbf{m})$ has no effect and the motion-corrupted network input results in a large loss as the network was trained on reconstructing motion-free data. Contrary, when the motion parameters are chosen correctly the network input is a motion-corrected ZF image, which is similar to a motion-free ZF image and results in a small loss. See Figure 2 for example images. In Section 5 below we study this loss theoretically.

We call this approach test-time-training, since the adjoint $\mathbf{A}^{\dagger}(\mathcal{T}, -\mathbf{m})$ can be considered to be part of the network, and by optimizing over the motion states $\mathbf{m}$ we are optimizing over part of the network's parameters. Methods that optimize a network at inference are referred to as test-time-training methods, and are successful at prediction under distribution shifts [39, 9].

In every iteration, the network reconstructs the entire 3D input volume slice-wise, where the slicing direction is sampled uniformly at random to be either in the $r_x \times r_y$, $r_x \times r_z$ or $r_y \times r_z$ image plane. We compute gradients only for a subset (of size 5, limited by GPU memory) of slices sampled independently in every iteration. While motion has a local effect in the k-space, the artifacts spread globally in the image space hence computing gradients with respect to a single slice can contain signal about all motion parameters.

In order to minimize the loss (4) reliably over different levels of motion severity the optimization scheme is important. We take a three-phase optimization approach.

Phase 1 optimizes over one motion state per acquired shot. We start with a large initial learning rate in order to explore the non-convex loss landscape. Especially for strong motion initializing all parameters with 0 can lead to a large distance to the true motion parameters. During phase 1 the learning rate is decayed twice in order to converge to a stable first estimate of the motion parameters.

At the start of phase 2 we compute the DC loss (4) for every estimated motion state $\hat{\mathbf{m}}_i$

$$\mathcal{L}_{\text{TTT}}(\hat{\mathbf{m}}_i) = \left\| \mathbf{A}(\mathcal{T}, \hat{\mathbf{m}}_i) f_{\hat{\boldsymbol{\theta}}} \left( \mathbf{A}^{\dagger}(\mathcal{T}, -\hat{\mathbf{m}})\mathbf{y} \right) - \mathbf{M}_{\hat{\mathbf{m}}_i}\mathbf{y} \right\|_1 / \|\mathbf{M}_{\hat{\mathbf{m}}_i}\mathbf{y}\|_1, \quad (5)$$

where the mask $\mathbf{M}_{\hat{\mathbf{m}}_i}$ keeps only the part of the k-space acquired during the $i$-th state. Motion states with a loss larger than a certain threshold are likely estimated poorly and we reset them to the average between the previous and next motion state that fall below the threshold. Phase 2 then only optimizes over the motion states that have been reset. For intra-shot motion, a thresholded motion state is not only reset, but $N_{\text{splits}}$ additional motion states up to the number of acquired k-space lines per shot can be introduced in order to estimate a more resolved motion trajectory for this shot.

Phase 3 again optimizes jointly over all motion states with a small learning rate to converge to a final estimate of the motion trajectory.

**Step 3: Reconstruction.** Using the estimated motion parameters $\hat{\mathbf{m}}$, we can obtain an estimate of the motion-corrected image directly from the network output $f_{\hat{\boldsymbol{\theta}}}(\mathbf{A}^{\dagger}(\mathcal{T}, -\hat{\mathbf{m}})\mathbf{y})$, and apply a data-consistency step to improve performance (*U-net-DCLayer*). This step moves the frequencies of the reconstructed image closer to the given frequencies, and is used for example by Chen et al. [4]. Alternatively, we can use any reconstruction method for motion-free data, such as a classical L1-minimization-based reconstruction [26]. We refer to L1 or U-net reconstruction based on motion parameters estimated by MotionTTT as *MotionTTT-L1* and *MotionTTT-U-net-DCLayer* and their respective performance with the oracle known motion parameters as *KnwonMotion-L1* and *KnownMotion-U-net-DCLayer*. We refer to the reconstruction after DC loss thresholding that excludes motion states with a large DC loss from the reconstruction as *MotionTTT+Th-L1*.

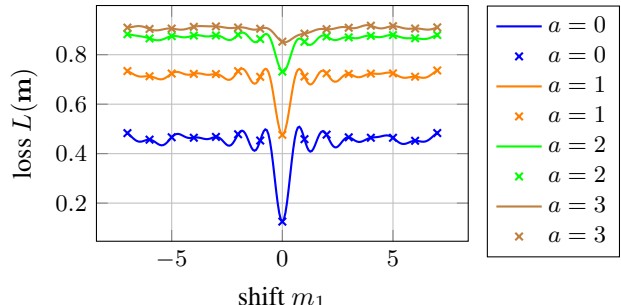

Figure 3: For an example with $n = 2k$, $k = 1400$, $d = 100$, and $b = 4$ and $\mathbf{m}^* = 0$ we plot the loss as a function of $m_1$, where $a$ is the number of values for $m_2, m_3, m_4$ that are set to an integer that is non-equal to $m_2^*, m_3^*, m_4^*$, respectively. It can be seen that there is a sharp minima around $\mathbf{m} = \mathbf{m}^*$. This minima turns out to be unique under certain conditions. In our theory we consider discrete shifts, indicated by crosses.

## 5 Theory for motion TTT

We consider the following model to illustrate the principle of our method. We consider a signal $\mathbf{x} \in \mathbb{R}^n$ the lies in a $d$ dimensional subspace described by the matrix $\mathbf{U} \in \mathbb{R}^{n \times d}$, i.e., there is a coefficient vector $\mathbf{c} \in \mathbb{R}^d$ so that $\mathbf{x} = \mathbf{U}\mathbf{c}$. We take the matrix $\mathbf{U}$ as a random Gaussian matrix with iid $\mathcal{N}(0, 1/\sqrt{n})$ entries, so that the columns of the matrix are approximately orthonormal.

Let $\mathbf{F}_{\mathcal{T}}$ be the Fourier matrix with rows chosen in the set $\mathcal{T} \subseteq \{0, \ldots, n-1\}$. We assume a measurement model where the signal $\mathbf{x}$ is shifted by unknown discrete integer parameters $m_1^*, \ldots, m_b^* \in \mathbb{Z}$, and for each shifted version of the signal, a set of measurements is collected according to

$$\mathbf{y}_\ell = \mathbf{D}_{m_\ell^*, \mathcal{T}_\ell} \mathbf{F}_{\mathcal{T}_\ell} \mathbf{x}, \tag{6}$$

where $\mathbf{D}_{m, \mathcal{T}_\ell}$ is a diagonal matrix with $e^{i2\pi m j/n}$, $j \in \mathcal{T}_\ell$ on its diagonal. Note that this multiplication with complex exponentials in the frequency domain implements a circular shift in the time domain. In this measurement model, the signal is assumed motion-free while the measurements in the set $\mathbf{F}_{\mathcal{T}_\ell}$ are collected. The frequencies in the set $\mathcal{T}_\ell$ are chosen by sampling each frequency independently with probability $k/n$. So in expectation, $k$ frequencies are included in the set $\mathcal{T}_\ell$.

We consider the network $f(\mathbf{x}) = \frac{n}{bk} \mathbf{U} \mathbf{U}^T \mathbf{x}$ for reconstructing a clean signal, where $bk$ is the total number of measurements collected. This choice of network is motivated by the fact that if the measurement is not motion corrupted, then we have that (see Appendix B.3)

$$f(\mathbf{F}_{\mathcal{T}}^* \mathbf{y}) \approx \mathbf{x}, \tag{7}$$

where $\mathcal{T} = \mathcal{T}_1 \cup \ldots \cup \mathcal{T}_b$ is the set of all measurements collected and $(\cdot)^*$ denotes the complex conjugate. We consider our test-time-training loss for this model, which is

$$L(\mathbf{m}) = \|\mathbf{D}_{\mathbf{m}} \mathbf{F}_{\mathcal{T}} f(\mathbf{F}_{\mathcal{T}}^* \mathbf{D}_{\mathbf{m}}^* \mathbf{y}) - \mathbf{y}\|_2^2. \tag{8}$$

Below, we show that for our model, under certain conditions and with high probability, the loss has a unique minimum at $L(\mathbf{m}^*)$. Before stating our result, we visualize the loss in Figure 3. It can be seen that the loss is not convex in $\mathbf{m}$ which makes it difficult to optimize.

**Theorem 1** *Consider the model introduced above, and assume that the signal $\mathbf{x} = \mathbf{U}\mathbf{c}$ is chosen randomly by drawing the entries of $\mathbf{c}$ iid from a zero-mean unit-variance Gaussian distribution. Let $a(\mathbf{m})$ be the number of values of $m_1, \ldots, m_b$ that are non-equal to $m_1^*, \ldots, m_b^*$. The following statement holds for all $\mathbf{m} \in \{0, \ldots, n-1\} \setminus \{\mathbf{m}^*\}$ simultaneously with high probability: If*

$$(1 - a(\mathbf{m})/b)^2 > c \frac{b^2 \log(n)^2 (b + d)}{n} \frac{n^2}{k^2 b^2} + c\sqrt{\frac{d}{bk}}, \tag{9}$$

*then $L(\mathbf{m}) > L(\mathbf{m}^*)$, where $c$ is a numerical constant.*

The theorem implies that if the subspace dimension, $d$, and the number of shifts, $b$, are sufficiently small relative to the number of measurements, $bk$, then the loss has a global minimum at the true shift $\mathbf{m}^*$.

# 6 Experiments

We demonstrate quantitatively and qualitatively on simulated data the ability of our method to accurately reconstruct images for a wide range of levels of motion severity in the presence of inter- and intra-shot motion. Moreover, on prospectively acquired real motion-corrupted data, we demonstrate that our method achieves significant gains in terms of visual reconstruction quality.

## 6.1 Simulated inter-shot motion experiments

We start with experiments with simulated inter-shot motion as described in Section 3. The model, data, and baselines considered for both inter- and intra-shot motion experiments are as follows.

**Model.** We use a 17.5M parameter U-net [32] $f_{\boldsymbol{\theta}}$, a common baseline with good performance on image-to-image tasks [46, 3, 13], training details are in Appendix D.

**Data.** We train the U-net $f_{\boldsymbol{\theta}}$ on motion-free data and evaluate MotionTTT with simulated motion-corrupted data sourced from the Calgary Campinas Brain MRI Dataset [37] (license CC BY-ND) consisting of 3D scans of size $k_x \times k_y \times k_z = 218 \times 170 \times 256$ and $C = 12$ receiver coils. We select 40 subjects for training from the training set, 4 subjects for validation and hyperparameter tuning, and 10 subjects for testing from the validation set. We train and test with an undersampling factor of 4 using the mask from Figure 1 (c). For training, we slice each 3D zero-filled reconstruction $\mathbf{A}^{\dagger}\mathbf{y}$ and the corresponding 3D reference volume $\mathbf{x}$ along all three dimensions resulting in about 25k pairs of 2D network inputs and targets. We compute sensitivity maps from $24 \times 24 \times 24$ auto-calibration lines of the originally fully-sampled motion-free k-space with ESPIRiT [41].

**Motion simulation.** We set the number of shots to $B = 50$ similar to how our own real data was acquired in the upcoming Section 6.3, and we use an interleaved sampling trajectory $\mathcal{T}$ (Figure 1 (c)), where every 50-th line in the k-space is acquired within one shot and the $3 \times 3$ center of the k-space containing the largest energy is sampled in the first shot. Without loss of generality, we assume the subject to be in zero-motion state at the first shot and hence do not simulate and estimate motion parameters for the first shot.

As the characteristics of patient motion vary widely, synthetic rigid body motion is often simulated as random motion with rotations and translations drawn uniformly from some range, or from a Gaussian [7, 22, 16, 34].

We simulate random motion with different levels of severity by varying the number of motion events $N_e \in \{1, 5, 10\}$ per scan and the maximum possible rotations/translations $M_{\max} \in \{2, 5, 10\}$ in degrees/mm. The shots between which a motion event occurs are sampled uniformly at random, and for each event translation and rotation parameters are sampled uniformly from $[-M_{\max}, M_{\max}]$. After a motion event the subject stays in this position until the next motion event occurs. This yields 10 different levels of motion severity including the motion free case.

**Baselines.** We compare to alternating optimization by Cordero-Grande et al. [7], which is one of very few approaches for retrospective motion estimation in 3D MRI. The method alternates between two steps of L1-minimization reconstruction with wavelet regularization while fixing the motion parameters and four steps of motion parameter estimation while fixing the reconstruction. For the final reconstruction we perform L1-minimization from scratch based on the estimated motion parameters (*AltOpt-L1*) and with additional DC loss thresholding based on the estimated motion parameters (*AltOpt+Th-L1*), to ensure a fair comparison to our method. We also perform L1-minimization without any motion estimation (*L1*). We further compare to the *E2E stacked U-net* with self-assisted priors [1], a method for end-to-end (E2E) motion artifact reduction in 3D MRI. The model from Al-Masni et al. [1] is trained with pairs of motion-free images and undersampled images corrupted by motion simulated as described above to predict the motion-corrected image with a single pass trough the network at inference.

Hyperparameters for MotionTTT and the baselines are in Appendix D.

**MotionTTT accurately estimates motion over a wide range of motion severities.** The results in Figure 4 show the reconstruction performance in PSNR as a function of motion severity averaged

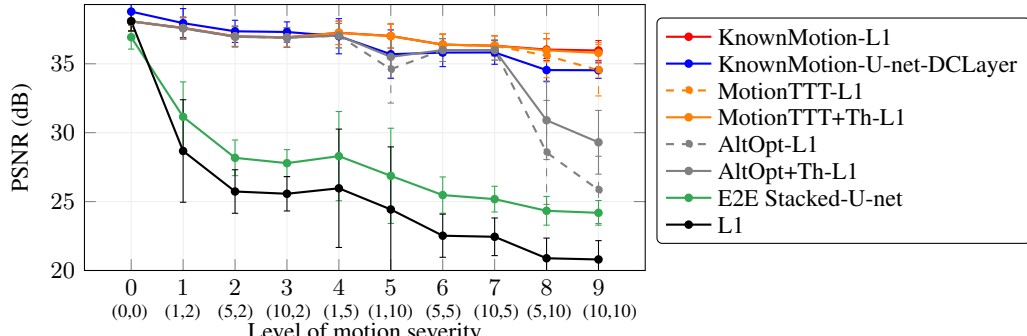

Figure 4: Reconstruction performance in PSNR as a function of the level of simulated *inter-shot motion* severity defined by (number of motion events, maximum rotation/translation in degrees/mm). We consider L1-minimization or U-net based reconstruction combined with either known motion, no motion-correction or motion estimated with MotionTTT or alternating optimization. Error bars are the standard deviation over test examples and randomly sampled motion trajectories.

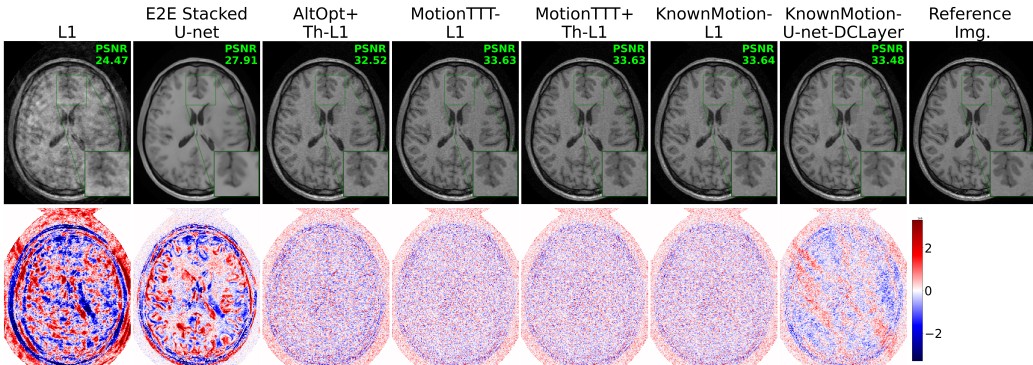

Figure 5: Reconstructions and difference images for simulated motion of severity level 5 for all methods in Figure 4.

over the test set and over two independently sampled motion trajectories per example. Figure 5 and Appendix E.1.1 contain example reconstructions. For both small and large motion severities, the PSNRs for reconstruction with known and estimated motion parameters is the same for MotionTTT, indicating that the motion parameters are estimated very well. Reconstruction results for the motion parameters itself are in Appendix E.1.1.

For *AltOpt* this is only true for small motion severities, for large ones MotionTTT significantly outperforms AltOpt. In addition, *MotionTTT* is about 6x faster than *AltOpt* for this problem, see Appendix E.1.2.

The performance of *E2E stacked U-net* lies significantly below the performance of methods that obtain a reconstruction based on explicitly estimating motion like *MotionTTT* or *AltOpt*. With increasing motion severity E2E methods suffer from the severe degradations in their network inputs, which can not be resolved without knowledge of the motion resulting in overly smooth reconstructions.

For the results in Figure 4 we used the fixed undersampling mask from Figure 1 (c) with acceleration factor 4. In Appendix E.1.3 we additionally ablate over acceleration factors 2 and 8 showing robust performance across acceleration factors and levels of motion severity.

**Reconstruction module.** Figure 4 shows that for no motion, U-net based reconstruction (*KnownMotion-U-net-DCLayer*) outperforms L1-based reconstruction (*KnownMotion-L1*), as expected, but for larger levels of motion severity L1-based reconstruction performs slightly better. A likely reason for this is a distribution shift, which is known to result in worse performance [8]: Acquiring MRI measurements under motion typically leads to some parts of the k-space being sampled

more than once, while parts that would be sampled if there were no motion are not sampled. This results in a change of effective undersampling mask and a motion-specific increase of the effective undersampling factor, which suggests why the reconstruction quality decays for all methods with increasing levels of motion even if motion parameters are known. This problem, illustrated in Figure 2, is well known [45, 12]. However, L1-based reconstruction is not as sensitive to changes in the mask as the U-net based reconstruction, which is trained with a fixed mask. Due to this distribution shift, the performance of U-net might degrade stronger than for L1-based reconstruction.

Finally, we note that, comparing *MotionTTT-L1* and *MotionTTT+Th-L1* Figure 4 shows that for strong motion, DC loss thresholding essentially closes the gap to *KnownMotion-L1*. This indicates that the thresholding does not unnecessarily exclude motion states for moderate motion, but reliably detects incorrectly estimated motion states for severe motion.

## 6.2   Simulated intra-shot motion experiments

Next, we study the performance of our method for intra-shot motion. As mentioned before, the model, data, and baselines are the same as for the inter-shot experiments in Section 6.1.

**Motion simulation.**   We again set the number of shots to $B = 50$. We use a random sampling trajectory (Figure 1 (d)), where the $3 \times 3$ center is acquired first and all other k-space lines are acquired at a random order, and again assume the subject to be in zero-motion state at the first shot. We randomly select $\lceil N_e/2 \rceil$ of the motion events to take place during the acquisition of one shot. Intra-shot motion is simulated by assigning a distinct motion state to each of the 182 k-space lines acquired during this shot such that the intra-shot motion trajectory connects the motion parameters from the previous to the next shot, where start and end point as well as the presence of up to two peaks due to over- and/or under-shooting within the intra-shot trajectory is randomized.

**Choice of the parameter $N_{\textbf{splits}}$.**   As explained in Section 4, MotionTTT estimates intra-shot motion by splitting motion states that exhibit a large DC loss after phase 1 of the optimization scheme into $N_{\text{splits}}$ motion states. Since ground truth intra-shot motion is simulated with a distinct motion state for each of the 182 k-space lines acquired during one shot, choosing the parameter $N_{\text{splits}} < 182$ results in a discretization error. On the other hand, choosing $N_{\text{splits}}$ large increases the difficulty of the estimation problem because the number of k-space lines corresponding to one motion state, decreases. We found $N_{\text{splits}} = 10$, i.e., about 18 k-space lines per motion state, to be a good trade-off, see Appendix E.2.1 for an ablation study.

**Sampling order.**   The sampling order within a shot is crucial for the ability to estimate intra-shot motion as estimating a motion state corresponding to a batch consisting of only high-frequency and low-energy components is difficult. In Appendix E.2.2 we ablate over different sampling orders and find that acquiring all k-space lines at a random order works well.

**MotionTTT with intra-shot motion estimation improves performance over discarding measurements corrupted by intra-shot motion.**   Figure 6 shows the reconstruction performance as a function of motion severity, where half of the motion events exhibit intra-shot motion. Recall that after phase 1 of the optimization scheme outlined in Section 4 MotionTTT converged to one estimated motion state per shot. *MotionTTT-L1* (Phase 1) is based on those motion states and hence does not estimate intra-shot motion. Consequently, even for the lowest level of motion considered here, *MotionTTT+Th-L1* (Phase 1) improves the performance as shots corrupted by intra-shot motion are discarded during DC loss thresholding before reconstruction.

In contrast, *MotionTTT-L1* (Phase 3) achieves on par performance with *MotionTTT+Th-L1* (Phase 3) for moderate motion levels indicating that intra-shot motion has been estimated successfully so that motion states are below the threshold we set for DC loss thresholding. Nevertheless, a performance gap remains relative to *KnownMotion-L1*, which results from the irreducible discretization error.

For severe motion the gap between *MotionTTT-L1* (Phase 3) and *MotionTTT+Th-L1* (Phase 3) increases as more motion states are estimated incorrectly and have to be discarded. However, MotionTTT with only Phase 1 continues to be outperformed indicating the benefit of performing intra-shot motion estimation over discarding measurements corrupted by intra-shot motion.

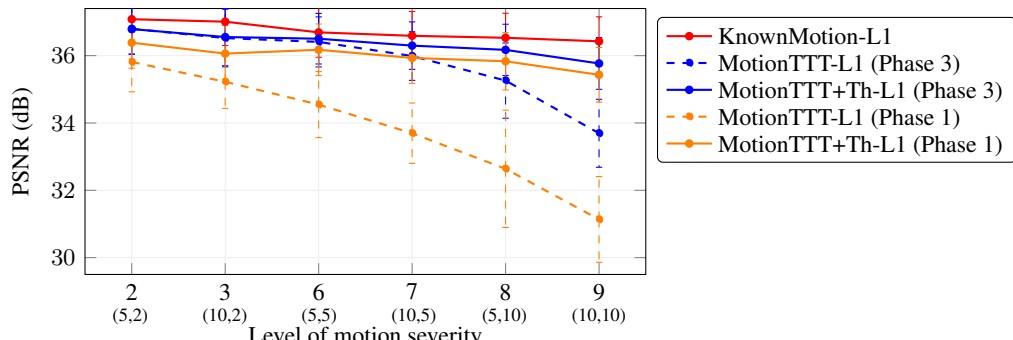

Figure 6: Reconstruction performance in PSNR as a function of the simulated motion severity defined as in Figure 4 only that here half of the motion events exhibit *intra-shot motion*. We consider L1-minimization based on known motion, motion estimated with MotionTTT after phase 1 (no intra-shot motion estimation) and after phase 3 (with intra-shot motion estimation) of the optimization scheme. Error bars are the standard deviation over test examples and randomly sampled motion trajectories.

Figure 11 in Appendix E contains a qualitative comparison, where especially in the case without DC loss thresholding the difference in reconstruction quality between phase 1 and 3 is clearly visible. After thresholding the differences are less visible, however that depends on the amount of intra-shot motion that we simulate as with more intra-shot motion the difference in undersampling factor between discarding and estimating intra-shot motion becomes larger and differences more visible.

### 6.3 Experiments with real motion

We now apply MotionTTT to real motion data. We use pre-trained model $f_\theta$ from Section 6.1, and the implementation details are described in Appendix D.

**Data.** We acquired four scans from one subject. To obtain motion-free reference data, mildly motion-corrupted, and strongly motion-corrupted data the subject was instructed to not move at all or move 1-3 times at distinct time points during the acquisition. The performed motions include nodding, head rotations, and either with or without returning to the original position. Sequence parameters (see Appendix D.4) were chosen to match those in the Calgary Campinas Brain MRI Dataset [37] as close as possible. Data was acquired with an undersampling factor of 4.94 and a random sampling trajectory (similar to Figure 1 (d)) with $B = 52$ shots. The acquisition of one shot lasts 1.3s followed by a pause of 1.6s resulting in a total scan duration of about 150s.

**Results.** We find that MotionTTT achieves significantly improved visual reconstruction quality compared to no motion-correction. Applying MotionTTT results in a significant reduction of motion artifacts for both mild and strong motion (reconstructed images are in Appendix E.3).

## 7 Limitations and future work

In this paper, we proposed the first deep-learning-based 3D rigid motion estimation method for 3D MRI and have demonstrated that it is effective and computationally managable at estimating motion and correcting for it.

As discussed in Section 6.1 our reconstruction module (Step 3) has room for improvement. We currently use L1-minimization, but a deep-learning method, e.g., regularization with diffusion models should yield further improvements.

Finally, the model used to perform MotionTTT can be improved in principle. Currently it is trained on fully-sampled data, which is scarce especially for 3D at high resolutions. It might be possible to train this module well with self-supervised training losses that require only undersampled data [44, 28, 20]. Moreover, as mentioned, there is a distribution shift in the mask and networks that work well with such distribution shifts might yield improvements.

## Acknowledgments and Disclosure of Funding

The authors are supported by the German Research Foundation (DFG) under grant numbers 456465471, 464123524, and 517586365.

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

## A   Computational aspects of simulating motion in the image or k-space

As mentioned in the Problem Statement Section 3, we simulate motion in the k-space rather than in the image domain because it is computationally more efficient if the number of motion states $b$ is larger than the number of coils $C$. Here, we elaborate on this statement.

In our work the MRI forward model under motion is implemented with the NUFFT, which for each shot in the sampling trajectory first computes the rotated coordinates based on the k-space data and the motion parameters of this shot. Then the k-space values at those coordinates for all shots can be obtained from a single application of the NUFFT. This requires us to compute a single interpolated version of the k-space data, which consists of $C$-many 3D coil volumes, thus $C$-many interpolated volumes need to be computed.

In contrast, simulating motion in the image domain required computing a transformed 3D image volume for each motion state, which then is expanded to the coil dimension and transformed to the k-space with the forward model (1). Hence, $b$ many interpolated volumes need to be computed. As in our work the number of coils $C = 12$ is smaller than the number of motion states $b$ it is computationaly more efficient to simulate motion in the k-space than in the image space.

## B   Proof of Theorem 1

In this appendix, we prove Theorem 1 from the theory Section 5. To prove the result, we upper bound the loss for the correct motion parameters, $L(\mathbf{m}^*)$ and lower bound the loss $L(\mathbf{m})$ for all other motion parameters $\mathbf{m} \neq \mathbf{m}^*$.

Assume without loss of generality that the ground-truth shift is equal to $\mathbf{m}^* = \mathbf{0}$. We have that

$$
L(\mathbf{m}) = \left\| \mathbf{D_m} \mathbf{F}_{\mathcal{T}} f(\mathbf{F}_{\mathcal{T}}^* \mathbf{D_m^*} \mathbf{y}) - \mathbf{y} \right\|_2^2
$$

$$
= \left\| \mathbf{D_m} \mathbf{F}_{\mathcal{T}} \frac{n}{bk} \mathbf{U} \mathbf{U}^T \mathbf{F}_{\mathcal{T}}^* \mathbf{D_m^*} \mathbf{D_{m^*}} \mathbf{F}_{\mathcal{T}} \mathbf{x} - \mathbf{D_{m^*}} \mathbf{F}_{\mathcal{T}} \mathbf{x} \right\|_2^2
$$

$$
\overset{\text{i}}{=} \left\| \frac{n}{bk} \mathbf{D_m} \mathbf{F}_{\mathcal{T}} \mathbf{U} \mathbf{U}^T \mathbf{F}_{\mathcal{T}}^* \mathbf{D_m^*} \mathbf{F}_{\mathcal{T}} \mathbf{x} - \mathbf{F}_{\mathcal{T}} \mathbf{x} \right\|_2^2
$$

$$
\overset{\text{ii}}{=} \left\| \frac{n}{bk} \mathbf{F}_{\mathcal{T}} \mathbf{U} \mathbf{U}^T \mathbf{F}^* \mathbf{D_m^*} \mathbf{F}_{\mathcal{T}} \mathbf{x} - \mathbf{D_m^*} \mathbf{F}_{\mathcal{T}} \mathbf{x} \right\|_2^2
$$

$$
= \left\| \frac{n}{bk} \mathbf{F}_{\mathcal{T}} \mathbf{U} \mathbf{U}^T \mathbf{F}_{\mathcal{T}}^* \mathbf{D_m^*} \mathbf{F}_{\mathcal{T}} \mathbf{U} \mathbf{c} - \mathbf{D_m^*} \mathbf{F}_{\mathcal{T}} \mathbf{U} \mathbf{c} \right\|_2^2,
$$

where $\mathbf{D}^*$ is the Hermitian transpose of the matrix $\mathbf{D}$. Here, equation i follows from the assumption that the optimal motion parameters are zero, thus $\mathbf{D_{m^*}} = \mathbf{I}$, and equation ii follows from the entries of $\mathbf{D_m}$ having absolute value one, and $\mathbf{D_m} \mathbf{D_m^*} = \mathbf{I}$.

We first upper bound $L(\mathbf{m}^*) = L(\mathbf{0})$. We have that

$$
L(\mathbf{0}) = \left\| \frac{n}{bk} \mathbf{F}_{\mathcal{T}} \mathbf{U} \mathbf{U}^T \mathbf{F}_{\mathcal{T}}^* \mathbf{F}_{\mathcal{T}} \mathbf{U} \mathbf{c} - \mathbf{F}_{\mathcal{T}} \mathbf{U} \mathbf{c} \right\|_2^2
$$

$$
\leq \left\| \frac{n}{bk} \mathbf{F}_{\mathcal{T}} \mathbf{U} \mathbf{U}^T \mathbf{F}_{\mathcal{T}}^* - \mathbf{I} \right\|^2 \left\| \mathbf{F}_{\mathcal{T}} \mathbf{U} \mathbf{c} \right\|_2^2
$$

$$
\leq \left( \frac{n}{bk} \sigma_{\max}^2(\mathbf{F}_{\mathcal{T}} \mathbf{U}) - 1 \right) \left\| \mathbf{F}_{\mathcal{T}} \mathbf{U} \mathbf{c} \right\|_2^2
$$

$$
\leq \left( \left( 1 + 2\sqrt{\frac{d}{bk}} \right)^2 - 1 \right) \frac{bk}{n} \left( 1 + 2\sqrt{\frac{d}{bk}} \right)^2 2 \tag{10}
$$

$$
\leq 3\sqrt{\frac{d}{bk}} \frac{bk}{n} 4. \tag{11}
$$

For the last inequality, we used that $d/bk \leq 1/4$ by assumption. According to Theorem 2.6 in Rudelson and Vershynin [33], the second to last inequality holds on the events

$$
\mathcal{E}_1 = \left\{ \sqrt{\frac{bk}{n}} \left( 1 - 2\sqrt{\frac{d}{bk}} \right) \leq \sigma_{\min}(\mathbf{F}_{\mathcal{T}} \mathbf{U}) \leq \sigma_{\max}(\mathbf{F}_{\mathcal{T}} \mathbf{U}) \leq \sqrt{\frac{bk}{n}} \left( 1 + 2\sqrt{\frac{d}{bk}} \right) \right\} \tag{12}
$$

and

$$\mathcal{E}_2 = \left\{ \left| \|\mathbf{c}\|_2 - 1 \right| \leq \frac{\beta}{\sqrt{n}} \right\},$$ (13)

with $\beta = \sqrt{n}$. Those events hold with the probabilities, for all $\beta > 0$

$$P[\mathcal{E}_1] \geq 1 - 2e^{-d/2},$$ (14)

$$P[\mathcal{E}_2] \geq 1 - 2e^{-c\beta^2}.$$ (15)

Next, we lower-bound $L(\mathbf{m})$ for $\mathbf{m} \neq \mathbf{0}$. Let $a$ be the number of individual motion parameters in the vector $\mathbf{m}$ that are non-equal to the true motion parameters $\mathbf{m}^* = \mathbf{0}$. We have

$$L(\mathbf{m}) = \left\| \mathbf{F}_{\mathcal{T}} \mathbf{U} \left( \frac{n}{bk} \mathbf{U}^T \mathbf{F}_{\mathcal{T}}^* \mathbf{D}_{\mathbf{m}}^* \mathbf{F}_{\mathcal{T}} \mathbf{U} - \frac{a}{b} \mathbf{I} \right) \mathbf{c} + \left( \mathbf{F}_{\mathcal{T}} \mathbf{U} \frac{a}{b} - \mathbf{D}_{\mathbf{m}}^* \mathbf{F}_{\mathcal{T}} \mathbf{U} \right) \mathbf{c} \right\|_2^2$$

$$\geq \left( \left\| \left( \mathbf{F}_{\mathcal{T}} \mathbf{U} \frac{a}{b} - \mathbf{D}_{\mathbf{m}}^* \mathbf{F}_{\mathcal{T}} \mathbf{U} \right) \mathbf{c} \right\|_2 - \left\| \mathbf{F}_{\mathcal{T}} \mathbf{U} \left( \frac{n}{bk} \mathbf{U}^T \mathbf{F}_{\mathcal{T}}^* \mathbf{D}_{\mathbf{m}}^* \mathbf{F}_{\mathcal{T}} \mathbf{U} - \frac{a}{b} \mathbf{I} \right) \mathbf{c} \right\|_2 \right)^2$$

$$\geq \left( \left\| \left( \mathbf{F}_{\mathcal{T}} \mathbf{U} \frac{a}{b} - \mathbf{D}_{\mathbf{m}}^* \mathbf{F}_{\mathcal{T}} \mathbf{U} \right) \mathbf{c} \right\|_2 - \left\| \mathbf{F}_{\mathcal{T}} \mathbf{U} \right\| \left\| \left( \frac{n}{bk} \mathbf{U}^T \mathbf{F}_{\mathcal{T}}^* \mathbf{D}_{\mathbf{m}}^* \mathbf{F}_{\mathcal{T}} \mathbf{U} - \frac{a}{b} \mathbf{I} \right) \mathbf{c} \right\|_2 \right)^2$$

$$\geq \left\| \left( \mathbf{F}_{\mathcal{T}} \mathbf{U} \frac{a}{b} - \mathbf{D}_{\mathbf{m}}^* \mathbf{F}_{\mathcal{T}} \mathbf{U} \right) \mathbf{c} \right\|_2^2 - \left\| \mathbf{F}_{\mathcal{T}} \mathbf{U} \right\|^2 \left\| \left( \frac{n}{bk} \mathbf{U}^T \mathbf{F}_{\mathcal{T}}^* \mathbf{D}_{\mathbf{m}}^* \mathbf{F}_{\mathcal{T}} \mathbf{U} - \frac{a}{b} \mathbf{I} \right) \mathbf{c} \right\|_2^2$$

$$\geq |1 - a/b|^2 \frac{bk}{n} \frac{1}{2} - \frac{7}{8} \frac{bk}{n} \cdot (1 + \alpha) \frac{\beta^2}{n} \frac{n^2}{k^2 b^2} (4b + d)$$

$$= |1 - a/b|^2 \frac{bk}{n} \frac{1}{2} - \frac{7}{8} \cdot (1 + \alpha) \frac{\beta^2}{n} \frac{n}{kb} (4b + d),$$

where the last inequality holds with probability at least $1 - e^{-c\alpha} + 2d(d + b)e^{-c\beta^2} - 3e^{-cd}$ which follows from the bounds

$$P\left[ \left\| \left( \frac{n}{bk} \mathbf{U}^T \mathbf{F}_{\mathcal{T}}^* \mathbf{D}_{\mathbf{m}}^* \mathbf{F}_{\mathcal{T}} \mathbf{U} - \frac{a}{b} \mathbf{I} \right) \mathbf{c} \right\|_2^2 \geq (1 + \alpha) \frac{\beta^2}{n} \frac{n^2}{k^2 b^2} (4b + d) \right] \leq e^{-c\alpha} + 2d(d + b)e^{-c\beta^2}$$ (16)

and

$$P\left[ \left\| \left( \mathbf{F}_{\mathcal{T}} \mathbf{U} \frac{a}{b} - \mathbf{D}_{\mathbf{m}}^* \mathbf{F}_{\mathcal{T}} \mathbf{U} \right) \mathbf{c} \right\|_2 \leq |1 - a/b| \sqrt{\frac{bk}{n} \frac{7^2}{8^2}} \right] \leq 3e^{-cd}.$$ (17)

Thus, we have with probability at least $1 - e^{-c\alpha} + 2d(d + b)e^{-c\beta^2} - 3e^{-cd} - 4e^{-cd}$ that $L(\mathbf{m}) > L(\mathbf{m}^*) = L(0)$ if

$$|1 - a/b|^2 \frac{bk}{n} \frac{1}{2} - \frac{7}{8} \frac{bk}{n} \cdot (1 + \alpha) \frac{\beta^2}{n} \frac{n^2}{k^2 b^2} (4b + d) - 12 \sqrt{\frac{d}{bk} \frac{bk}{n}} > 0$$ (18)

which is equivalent to

$$|1 - a/b|^2 > \frac{7}{4} \cdot (1 + \alpha) \frac{\beta^2}{n} \frac{n^2}{k^2 b^2} (4b + d) + 24 \sqrt{\frac{d}{bk}}.$$ (19)

By a union bound over all motion parameter $\mathbf{m} \in \{0, \ldots, n - 1\} \setminus \{\mathbf{m}^*\}$ (there are $n^b$ many), we have that

$$P\left[ \max_{\mathbf{m} \neq \mathbf{m}^*} L(\mathbf{m}) \geq L(\mathbf{m}^*) \right] \leq \sum_{\mathbf{m} \neq \mathbf{m}^*} P[L(\mathbf{m}) \geq L(\mathbf{m}^*)]$$

$$\leq n^b \left( e^{-c\alpha} + 2d(d + b)e^{-c\beta^2} - 7e^{-cd} \right)$$

$$\leq \left( e^{-c} + e^{-c} - 7e^{-cd - c' \log(n)b} \right)$$

provided that

$$|1 - a/b|^2 > \frac{7}{4} \cdot b \log(n) \frac{b \log(n)}{n} \frac{n^2}{k^2 b^2} (4b + d) + 24 \sqrt{\frac{d}{bk}},$$ (20)

which concludes the proof. It remains to prove the intermediate results, which we do next.

## B.1 Lower bounding the first term, proof of equation (17):

Since the entries of $\mathbf{D}_{\mathbf{m}}^*$ have absolute value one, and $a/b \in [0,1]$, we have

$$\left\| \left( \mathbf{F}_{\mathcal{T}}\mathbf{U}\frac{a}{b} - \mathbf{D}_{\mathbf{m}}^*\mathbf{F}_{\mathcal{T}}\mathbf{U} \right)\mathbf{c} \right\|_2 \geq \|(1 - a/b)\mathbf{F}_{\mathcal{T}}\mathbf{U}\mathbf{c}\|_2$$

$$\geq |1 - a/b|\sigma_{\min}(\mathbf{F}_{\mathcal{T}}\mathbf{U})\|\mathbf{c}\|_2$$

$$\geq |1 - a/b|\sqrt{\frac{bk}{n}}\left(1 - \sqrt{\frac{d}{bk}}\right)\left(1 - \frac{\beta}{\sqrt{n}}\right)$$

$$\geq |1 - a/b|\sqrt{\frac{bk}{n}}\frac{7^2}{8^2},$$

where the second to last inequality holds with probability at least $1 - 3e^{-cd}$ according to equations (14) and (15), and where we used that $\frac{d}{bk} \leq 1/64$ and $\frac{d}{n} \leq \frac{1}{64}$.

## B.2 Lower-bounding the second term, proof of equation (16)

Define $\mathbf{A} = \frac{n}{bk}\mathbf{U}^T\mathbf{F}_{\mathcal{T}}^*\mathbf{D}_{\mathbf{m}}^*\mathbf{F}_{\mathcal{T}}\mathbf{U} - \frac{a}{b}\mathbf{I}$ for notational convenience. We have that

$$\mathrm{P}\left[\|\mathbf{A}\mathbf{c}\|_2^2 \geq (1 + \alpha)\frac{\beta^2}{n}\frac{n^2}{k^2b^2}(4b + d)\right]$$

$$\leq \mathrm{P}\left[\|\mathbf{A}\|_F^2 \geq d\frac{\beta^2}{n}\frac{n^2}{k^2b^2}(4b + d)\right] + \mathrm{P}\left[\|\mathbf{A}\mathbf{c}\|_2^2 \geq (1 + \alpha)\frac{1}{d}\|\mathbf{A}\|_F^2\right]$$

$$\leq \mathrm{P}\left[\|\mathbf{A}\|_F^2 \geq d\frac{\beta^2}{n}\frac{n^2}{k^2b^2}(4b + d)\right] + \mathrm{P}\left[\|\mathbf{A}\mathbf{c}\|_2^2 - \frac{1}{d}\|\mathbf{A}\|_F \geq \alpha\frac{1}{d}\|\mathbf{A}\|_F\right]$$

$$\leq e^{-c\alpha} + 2d(d + b)e^{-c\beta^2}$$

where the last inequality follows from the Hanson-Wright inequality as well as from

$$\mathrm{P}\left[\|\mathbf{A}\|_F^2 \geq d\frac{\beta^2}{n}\frac{n^2}{k^2b^2}(4b + d)\right] \leq d2(1 + b)e^{-c\beta^2} + d^22e^{-c\beta^2}. \tag{21}$$

We next prove the bound (21). We start with upper bounding the Frobenius norm of the matrix $\mathbf{A}$. We split the squared Frobenius norm into a sum of the squared diagonal entries and squared off-diagonal entries according to

$$\|\mathbf{A}\|_F^2 = \left\|\frac{n}{kb}\mathbf{U}^T\mathbf{F}_{\mathcal{T}}^*\mathbf{D}_{\mathbf{m}}^*\mathbf{F}_{\mathcal{T}}\mathbf{U} - \frac{a}{b}\mathbf{I}\right\|_F^2$$

$$= \sum_{i=1}^d \left(\frac{n}{kb}\mathbf{u}_i^*\mathbf{F}_{\mathcal{T}}^*\mathbf{D}_{\mathbf{m}}^*\mathbf{F}_{\mathcal{T}}\mathbf{u}_i - \frac{a}{b}\right)^2 + \sum_{i=1}^d\sum_{j\neq i}^d \left(\frac{n}{kb}\mathbf{u}_i^*\mathbf{F}_{\mathcal{T}}^*\mathbf{D}_{\mathbf{m}}^*\mathbf{F}_{\mathcal{T}}\mathbf{u}_j\right)^2.$$

It follows that

$$\mathrm{P}\left[\|\mathbf{A}\|_F^2 \geq d\frac{\beta^2}{n}\frac{n^2}{k^2b^2}(4b + d)\right] \leq \mathrm{P}\left[\|\mathbf{A}\|_F^2 \geq d\left(\frac{\beta}{\sqrt{n}}\left(1 + \frac{n}{k\sqrt{b}}\right)\right)^2 + d(d - 1)\left(\frac{n}{kb}\right)^2\frac{\beta^2}{n}\right]$$

$$\leq \sum_{i=1}^d \mathrm{P}\left[\left|\frac{n}{kb}\mathbf{u}_i^*\mathbf{F}_{\mathcal{T}}^*\mathbf{D}_{\mathbf{m}}^*\mathbf{F}_{\mathcal{T}}\mathbf{u}_i - \frac{a}{b}\right|^2 \geq \left(\frac{\beta}{\sqrt{n}}\left(1 + \frac{n}{k\sqrt{b}}\right)\right)^2\right]$$

$$+ \sum_{i=1}^d\sum_{j\neq i}^d \mathrm{P}\left[\left(\frac{n}{kb}\mathbf{u}_i^*\mathbf{F}_{\mathcal{T}}^*\mathbf{D}_{\mathbf{m}}^*\mathbf{F}_{\mathcal{T}}\mathbf{u}_j\right)^2 \geq \left(\frac{n}{kb}\right)^2\frac{\beta^2}{n}\right]$$

$$\leq d2(1 + b)e^{-c\beta^2} + d^2e^{-c\beta^2}.$$

The last inequality follows from

$$\mathrm{P}\left[\left|\frac{n}{kb}\mathbf{u}_i^*\mathbf{F}_{\mathcal{T}}^*\mathbf{D}_{\mathbf{m}}^*\mathbf{F}_{\mathcal{T}}\mathbf{u}_i - \frac{a}{b}\right| \geq \frac{\beta}{\sqrt{n}}\left(1 + \frac{n}{k\sqrt{b}}\right)\right] \leq 2(1 + b)e^{-c\beta^2}. \tag{22}$$

and

$$\mathrm{P}\left[|\mathbf{u}_i^*\mathbf{F}_{\mathcal{T}}^*\mathbf{D}_\mathbf{m}^*\mathbf{F}_{\mathcal{T}}\mathbf{u}_j| \geq \frac{\beta}{\sqrt{n}}\right] \leq 3e^{-c\beta^2}. \tag{23}$$

**Bounding a diagonal entry, proof of inequality** (22): First, consider a diagonal element and let $\mathbf{Z}_\ell \in \mathbb{R}^{n\times n}$ be the mask that selects the frequencies in the set $\mathcal{T}_\ell$, and recall that $\mathbf{F} \in \mathbb{R}^{n\times n}$ is the Fourier transform. With a slight abuse of notation, we let $\mathbf{D}_m$ be the diagonal matrix that contains the frequencies that matrix the Fourier matrix it is multiplied with, i.e., in $\mathbf{D}_m\mathbf{F}$, the matrix $\mathbf{D}_m$ is the $n \times n$ diagonal matrix with entries $e^{i2\pi m\ell/n}$, $\ell = 0, \ldots, n-1$, and in $\mathbf{D}_m\mathbf{F}_{\mathcal{T}}$, the matrix $\mathbf{D}_m$ is the $|\mathcal{T}| \times |\mathcal{T}|$ diagonal matrix with entries $e^{i2\pi m\ell/n}$, $\ell \in \mathcal{T}$. For convenience, we drop the index $i$ and write $\mathbf{u} = \mathbf{u}_i$. With this we have

$$
\begin{aligned}
\frac{n}{kb}\mathbf{u}^*\mathbf{F}_{\mathcal{T}}^*\mathbf{D}_\mathbf{m}^*\mathbf{F}_{\mathcal{T}}\mathbf{u} - \frac{a}{b} &= -\frac{a}{b} + \sum_{\ell=1}^b \frac{n}{kb}\mathbf{u}^*\mathbf{F}_{\mathcal{T}_\ell}^*\mathbf{D}_{m_\ell}^*\mathbf{F}_{\mathcal{T}_\ell}\mathbf{u} \\
&= -\frac{a}{b} + \sum_{\ell=1}^b \frac{n}{kb}\mathbf{u}^*\mathbf{F}^*\mathbf{Z}_\ell\mathbf{D}_{m_\ell}^*\mathbf{F}\mathbf{u} \\
&= -\frac{a}{b} + \sum_{\ell=1}^b \frac{n}{kb}\mathbf{u}^*\mathbf{F}^*\left(\mathbf{Z}_\ell - \frac{k}{n}\mathbf{I}\right)\mathbf{D}_{m_\ell}^*\mathbf{F}\mathbf{u} + \frac{1}{b}\mathbf{u}^*\mathbf{F}^*\mathbf{D}_{m_\ell}^*\mathbf{F}\mathbf{u} \\
&= \frac{n}{kb}\sum_{\ell=1}^b \tilde{\mathbf{u}}^*\left(\mathbf{Z}_\ell - \frac{k}{n}\mathbf{I}\right)\mathbf{D}_{m_\ell}^*\tilde{\mathbf{u}} + \left(\sum_{\ell=1}^b \frac{1}{b}\mathbf{u}^*\mathbf{F}^*\mathbf{D}_{m_\ell}^*\mathbf{F}\mathbf{u}\right) - \frac{a}{b}. \quad (24)
\end{aligned}
$$

Here, the entries of $\tilde{\mathbf{u}} = \mathbf{F}\mathbf{u} \in \mathbb{R}^n$ are iid $\mathcal{CN}(0, 1/\sqrt{n})$ distributed, since the DFT matrix $\mathbf{F}$ has orthonormal columns.

Thus, by a union bound

$$
\begin{aligned}
&\mathrm{P}\left[\left|\frac{n}{kb}\mathbf{u}_i^*\mathbf{F}_{\mathcal{T}}^*\mathbf{D}_\mathbf{m}^*\mathbf{F}_{\mathcal{T}}\mathbf{u}_i - \frac{a}{b}\right| \geq \frac{\beta}{\sqrt{n}}\left(1 + \frac{n}{k\sqrt{b}}\right)\right] \\
&\leq \mathrm{P}\left[\left|\frac{n}{kb}\sum_{\ell=1}^b \tilde{\mathbf{u}}^*\left(\mathbf{Z}_\ell - \frac{k}{n}\mathbf{I}\right)\mathbf{D}_{m_\ell}^*\tilde{\mathbf{u}}\right| \geq \frac{n}{kb}\frac{\sqrt{b}\beta}{\sqrt{n}}\right] + \mathrm{P}\left[\left(\sum_{\ell=1}^b \frac{1}{b}\mathbf{u}^*\mathbf{F}^*\mathbf{D}_{m_\ell}^*\mathbf{F}\mathbf{u}\right) - \frac{a}{b} \geq \frac{\beta}{\sqrt{n}}\right] \\
&\leq 2e^{-c\beta^2} + b2e^{-c\beta^2} = 2(1+b)e^{-c\beta^2}. \quad (25)
\end{aligned}
$$

where we used that, for all $\beta > 0$,

$$\mathrm{P}\left[\left|\sum_{\ell=1}^b \tilde{\mathbf{u}}^*\left(\mathbf{Z}_\ell - \frac{k}{n}\mathbf{I}\right)\mathbf{D}_{m_\ell}^*\tilde{\mathbf{u}}\right| \geq \frac{\sqrt{b}\beta}{\sqrt{n}}\right] \leq 2e^{-c\beta^2} \tag{26}$$

and

$$\mathrm{P}\left[\left(\sum_{\ell=1}^b \frac{1}{b}\mathbf{u}^*\mathbf{F}^*\mathbf{D}_{m_\ell}^*\mathbf{F}\mathbf{u}\right) - \frac{a}{b} \geq \frac{\beta}{\sqrt{n}}\right] \leq b2e^{-c\beta^2} \tag{27}$$

This concludes the proof of inequality (22). It remains to proof equations (26) and (27).

**Proof of equation** (26): Note that

$$\tilde{\mathbf{u}}^*\left(\mathbf{Z}_b - \frac{k}{n}\mathbf{I}\right)\mathbf{D}_b^*\tilde{\mathbf{u}} = \sum_{i=1}^n \tilde{u}_i^*\tilde{u}_i e^{i2\pi mi/n}\left(z_{b_i} - \frac{k}{n}\right). \tag{28}$$

Since $z_{b,i} - \frac{k}{n}$ is a sub-Gaussian zero mean random variable a concentration inequality for sub-Gaussians yields

$$\mathrm{P}\left[\left|\tilde{\mathbf{u}}^*\left(\mathbf{Z}_b - \frac{k}{n}\mathbf{I}\right)\mathbf{D}_b^*\tilde{\mathbf{u}}\right| \geq \frac{\beta}{\sqrt{n}}\right] \leq 2e^{-\frac{c(\beta/\sqrt{n})^2}{\|\tilde{\mathbf{u}}^*\cdot\tilde{\mathbf{u}}\|_2^2}} \leq 3e^{-c\beta^2}. \tag{29}$$

Here, we used that $\tilde{\mathbf{u}}^* \cdot \tilde{\mathbf{u}}$ is the entrywise product, and the random variable $\|\tilde{\mathbf{u}}^* \cdot \tilde{\mathbf{u}}\|_2^2 = \sum_{i=1}^{n} (\tilde{u}_i^* \tilde{u}_i)^2$ concentrates around its expectation $n 3 \sigma^4 = n 3/n^2 = 3/n$.

Thus we have shown that the random variables $s_\ell = \tilde{\mathbf{u}}^* \left( \mathbf{Z}_\ell - \frac{k}{n} \mathbf{I} \right) \mathbf{D}_b^* \tilde{\mathbf{u}}$ (conditioned on $\tilde{\mathbf{u}}$) are sub-Gaussian. The random variables are also zero-mean and independent (if conditioned on $\tilde{\mathbf{u}}$), and thus by concentration of sub-Gaussian random variables we get

$$
P \left[ \left| \sum_{\ell=1}^{b} \tilde{\mathbf{u}}^* \left( \mathbf{Z}_\ell - \frac{k}{n} \mathbf{I} \right) \mathbf{D}_{m_\ell}^* \tilde{\mathbf{u}} \right| \geq \frac{1}{\sqrt{n}} \beta \sqrt{b} \right] \leq 2 e^{-c\beta^2}.
$$

This concludes the proof of equation (26).

**Proof of equation** (27): Next, consider the random variable $\mathbf{u}^* \mathbf{F}^* \mathbf{D}_{m_\ell}^* \mathbf{F} \mathbf{u} = \tilde{\mathbf{u}}^* \mathbf{D}_{m_\ell}^* \tilde{\mathbf{u}}$ in equation (24). If $m_\ell = 0$, i.e., there is no shift, we have $\mathbf{D}_0 = \mathbf{I}$, and thus the random variable becomes $\tilde{\mathbf{u}}^* \tilde{\mathbf{u}}$ which is a sum of sub-exponential random variables and concentrates around 1. In case $m_\ell$ is an integer non-equal to zero, we have that $\mathbf{u}^* \mathbf{F}^* \mathbf{D}_{m_\ell}^* \mathbf{F} \mathbf{u} = \mathbf{u}^T \mathbf{u}_{m_\ell}$, where $\mathbf{u}_{m_\ell}$ is a vector circularly shifted by $m_\ell$. We can write $\mathbf{u}^T \mathbf{u}_{m_\ell}$ as two sums of independent Gaussians, to see this consider the case $m_\ell = 1$ and note that for this case we have

$$
\mathbf{u}^T \mathbf{u}_{m_\ell} = \underbrace{(u_1 u_2 + u_3 u_4 + \ldots)}_{S_1} + \underbrace{(u_2 u_3 + u_4 u_5 + \ldots)}_{S_2}.
$$

By the union bound and by Hoeffding's inequality, we get that

$$
P \left[ \mathbf{u}^T \mathbf{u}_{m_\ell} \geq 2\beta/\sqrt{n} \right] \leq P \left[ S_2 \geq \beta/\sqrt{n} \right] + P \left[ S_2 \geq \beta/\sqrt{n} \right] \leq 2 e^{-c\beta^2}. \tag{30}
$$

Combining this via a union bound for all summands $\ell = 1, \ldots, b$ yields the bound in equation (27).

**Bounding an off-diagonal element, proof of inequality** (23): Now consider an off-diagonal element $\tilde{\mathbf{u}}_i^* \mathbf{D}_{\mathbf{m}}^* \tilde{\mathbf{u}}_j$. Since $\tilde{\mathbf{u}}_i$ and $\tilde{\mathbf{u}}_j$ are independent and the entries have zero mean and variance $1/n$, this is a sum of Gaussian random variables with variance $1/n$. Thus, by sub-Gaussian concentration or Hoeffding's inequality, we have that

$$
P \left[ |\tilde{\mathbf{u}}_i^* \mathbf{D}_{\mathbf{m}}^* \tilde{\mathbf{u}}_j| \geq \frac{\beta}{\sqrt{n}} \right] \leq 3 e^{-\frac{c\beta^2}{\|\mathbf{D}_{\mathbf{m}}^* \tilde{\mathbf{u}}_j\|_2^2}} = 2 e^{-c\beta^2}, \tag{31}
$$

where we used that $\|\mathbf{D}_{\mathbf{m}}^* \tilde{\mathbf{u}}_j\|_2^2 = \|\tilde{\mathbf{u}}_j\|_2^2$ concentrates around 1.

### B.3 Comment on Equation (7):

In the main body, we stated Equation (7), i.e.,

$$
f(\mathbf{F}_{\mathcal{T}}^* \mathbf{y}) \approx \mathbf{x}, \tag{32}
$$

where $\mathcal{T} = \mathcal{T}_1 \cup \ldots \cup \mathcal{T}_b$ is the set of all measurements collected.

To see that this approximation is accurate, note that for the noiseless case with a known shift, where $\mathbf{y} = \mathbf{D}_{\mathbf{m}} \mathbf{U} \mathbf{c}$ the network approximately reconstructs the signal since

$$
f((\mathbf{D}_{\mathbf{m}} \mathbf{F}_{\mathcal{T}})^\dagger \mathbf{y}) = f(\mathbf{F}_{\mathcal{T}}^* \mathbf{D}_{\mathbf{m}}^* \mathbf{y}) \tag{33}
$$
$$
= \mathbf{U} \mathbf{U}^T \mathbf{F}_{\mathcal{T}}^* \mathbf{D}_{\mathbf{m}}^* \mathbf{D}_{\mathbf{m}} \mathbf{F}_{\mathcal{T}} \mathbf{U} \mathbf{c} \tag{34}
$$
$$
= \mathbf{U} \mathbf{U}^T \mathbf{F}_{\mathcal{T}}^* \mathbf{F}_{\mathcal{T}} \mathbf{U} \mathbf{c} \tag{35}
$$
$$
\approx \mathbf{U} \mathbf{c} = \mathbf{x}, \tag{36}
$$

where the first equality follows because the matrix $\mathbf{D}_{\mathbf{m}} \mathbf{F}_{\mathcal{T}}$ has orthonormal rows, and the approximation holds since $\mathbf{U}^T \mathbf{F}_{\mathcal{T}}^* \mathbf{F}_{\mathcal{T}} \mathbf{U}$ concentrates around $\frac{n}{kb}$ if $\mathbf{U}$ is a random subspace, and if the number of measurements, $\mathcal{T}$ is sufficiently large relative to the dimension of the subspace, $d$.

# C  Ablation Study on Pre-training Loss

In Section 4, we stated that using a training loss that consists of the two losses, one computed in the image domain and one in the measurement domain, is beneficial over using only a loss in the measurement domain. In this section, we conduct the corresponding ablation study. We evaluate three distinct loss functions: image domain loss, k-space loss, and a combined loss. The image domain loss is as follows:

$$\mathcal{L}_{\text{train}}(\boldsymbol{\theta}) = \sum_{i=1}^{N} \left\| f_{\boldsymbol{\theta}}(\mathbf{A}^{\dagger}\mathbf{y}_i) - |\mathbf{x}_i| \right\|_1 / \|\mathbf{x}_i\|_1, \tag{37}$$

and the k-space loss is

$$\mathcal{L}_{\text{train}}(\boldsymbol{\theta}) = \sum_{i=1}^{N} \left\| \mathbf{F}\mathbf{E} f_{\boldsymbol{\theta}}(\mathbf{A}^{\dagger}\mathbf{y}_i) - \mathbf{F}\mathbf{E}\mathbf{x}_i \right\|_1 / \|\mathbf{F}\mathbf{E}\mathbf{x}_i\|_1. \tag{38}$$

To compare which of two loss functions or the combination of them is best, we trained three U-Net models with each loss function. All models were trained under identical settings, as detailed in Appendix D, with the exception that the model trained with magnitude loss had a single output layer representing the magnitude of the MRI image.

We evaluate the performance of the models in terms of their reconstruction quality as well as as a component of MotionTTT. First, we measure the reconstruction quality on motion free data. Second, we apply the U-Net within the *MotionTTT-Th-L1* framework on data with interleaved inter-shot motion at severity level 9, following the same setup as described in Section 6.1. Due to the requirement for a complex-valued U-Net in the MotionTTT framework, the magnitude-only U-Net can not be used within MotionTTT. The results are presented in the table below and show that the combined loss function provides the best performance for both motion-free reconstructions and when used within the *MotionTTT-Th-L1* framework for correcting motion-corrupted data.

| U-Net Training Loss | Image Domain Loss | k-space Loss | Combined Loss |
|---|---|---|---|
| Motion-free PSNR | 36.64 | 36.59 | 36.73 |
| *MotionTTT-Th-L1* (Severity 9) PSNR | Not Applicable | 35.23261 | 35.23587 |

# D  Hyperparameter configurations and implementation details

In this section we provide additional information for the experiments presented in Section 6 including hyperparameter configurations and implementation details for our proposed MotionTTT and the baselines alternating optimization and E2E stacked U-net and the MRI sequence parameters used to acquire the data for our real motion experiments.

## D.1  MotionTTT - Implementation details

In this section we discuss implementation details for the three components of MotionTTT pre-training, test-time-training for motion estimation and reconstruction together with some computational aspects.

### D.1.1  Pre-training

We train the standard U-net from the fastMRI repository [46] (MIT license) with 48 channels in the first layer and 4 blocks in the down-/up-sampling part resulting in 17.5M network parameters. The real and imaginary parts of the complex valued input and output images are processed in two network channels. We train for 240 step with the Adam optimizer with learning rate 0.001 which is decayed once by a factor of 10 after 200 steps. In every step one of the 40 training volumes is loaded and in each plane $r_x \times r_y$, $r_x \times r_z$ and $r_y \times r_z$ 20 random slices are backpropagated in three separate batches, i.e., 3 gradient steps per step. The model was trained for 17h on a Nvidia RTX A6000 GPU.

Throughout (pretraining and test-time-training) the sensitivity maps for the experiments with simulated motion are compute from $24 \times 24 \times 24$ auto-calibration lines of the originally fully-sampled motion-free k-space using ESPIRiT [41] with the BART toolbox.

### D.1.2 Test-time-training for motion estimation

To perform motion parameter estimation as outlined in Section 4 we use gradient based optimization with the Adam [19] optimizer. Phase 1 runs for 70 iterations with an initial learning rate of 4.0, which is decayed twice by a factor of 4 at iterations 40 and 60. The DC loss threshold used to determine incorrectly estimated motion states after phase 1 is set to 0.575. If no motion states fall above the threshold, the optimization continues for another 30 iterations with one additional learning rate decay after 10 iterations. If motion states fall above the threshold, phase 2 runs for 30 iterations with a learning rate of 0.5. Finally, phase 3 runs for another 30 iterations with a leraning rate of 0.05.

For severe motion we found that optimizing only over rotation parameters for the first few steps facilitates their correct estimation. To avoid single motion parameters to get stuck at a large value early during the optimization we clamp the estimated motion parameters at [5,8,10,12,15] degrees/millimeters for steps smaller than [15,30,45,60,150].

We use `TorchKbNufft` (MIT license), an implementation of the NUFFT from Muckley et al. [29] and use the option for density compensation based on the method of Pipe [31] when applying the adjoint NUFFT. To allow differentiation with respect to the input coordinates we build on the extension Bindings-NUFFT-pytorch from Alban Gossard (MIT license).

### D.1.3 Computational aspects

We run MotionTTT on a Nvidia L40 GPU with 46GB memory or on a Nvidia A100 GPU with 80GB memory. Within our implementation there are three hyperparameters that control the required GPU memory. Recall that in every iteration of MotionTTT the entire 3D volume is reconstructed slice-wise. However, gradients are only computed for a subset of randomly selected slices of size 5 as described in Section 4. The size of this subset is the first hyperparameter that we can control.

The second parameter is the batch size of the NUFFT. As described in Appendix A the NUFFT is applied for each of the $C$ coils. The `TorchKbNufft` package allows batch wise computation at the cost of increased GPU memory utilization.

As in our 3D setup we have to deal with a lot more motion states ($B = 50$) than previous work that studied the 2D setup, we implemented a third option to reduce GPU memory consumption. To this end, we split the estimated motion states into two batches of size of, e.g., 25 each and backpropagate the gradients subsequently before performing a single optimizer step that updates the estimated motion states.

Note that hyperparameters batch size of NUFFT and of motion states per backpropagation only affect the run time but do not change the optimization problem, where the cost of the latter could be compensated by increasing the number of GPUs accordingly. Hence, at the cost of prolonged run times or a larger GPU cluster MotionTTT can be applied with any number of motion states. With our hardware and for our $C = 12$-many coils we could use a NUFFT batch size of 12 (4) and a batch of 50 (25) motion states for backpropagation in case of the A100 (L40) GPU.

### D.1.4 Reconstruction

We perform DC loss thresholding for excluding the k-space data acquired during the $i$-th shot from the reconstruction based on its estimated motion state $\hat{\mathbf{m}}_i$ and the DC loss (5) $\mathcal{L}_{\text{TTT}}(\hat{\mathbf{m}}_i) > \delta$ with a threshold of $\delta = 0.575$.

We perform L1-minimization with wavelet regularization based on the estimated motion parameters. We run 50 steps with SGD and a learning rate of $5 \times 10^7$ and regularization weight $\lambda = 10^{-3}$.

The U-Net reconstruction given the true motion parameters *KnownMotion-U-net-DCLayer* from Figure 4, is obtained by fine-tuning a DCLayer as proposed in Chen et al. [4]. If the U-Net reconstruction is $\mathbf{x}_{\text{U-Net}} = f_{\hat{\boldsymbol{\theta}}}\left(\mathbf{A}^\dagger(\mathcal{T}, -\hat{\mathbf{m}})\mathbf{y}\right)$, then the reconstruction of the DCLayer is:

$$\hat{\mathbf{x}} = \arg\min_{\mathbf{x}} \frac{\|\mathbf{A}(\mathcal{T}, \hat{\mathbf{m}})\mathbf{x} - \mathbf{y}\|_1}{\|\mathbf{y}\|_1} + \lambda \frac{\|\mathbf{x} - \mathbf{x}_{\text{U-Net}}\|_1}{\|\mathbf{x}_{\text{U-Net}}\|_1}$$

In the experiments, we set $\lambda = 0.1$. The choice of learning rate and number of steps is critical for optimizing the DCLayer. After a grid searching, we identified the optimal learning rate and steps for various severity levels, which are summarized in the following table:

| Severity Level | 0, 1 | 2, 3, 4, 5 | 6, 7 | 8, 9 |
|---|---|---|---|---|
| Learning Rate | $1 \times 10^{10}$ | $1 \times 10^{10}$ | $1 \times 10^{10}$ | $5 \times 10^{10}$ |
| Steps | 20 | 50 | 100 | 50 |

### D.1.5 Hyperparameters for real motion experiments

For applying MotionTTT to the real motion-corrupted data in Section 6.3 we use the same hyper-parameter configuration as described above up to choosing a slightly smaller initial learning rate of 1.0 during phase 1 to explore the space of motion parameters more slowly and set the threshold parameter to 0.70 due to a generally higher DC loss level of the scanner data compared to the data used for the simulation.

For obtaining the final reconstruction with L1-minimization based on the motion parameters estimated with MotionTTT we set the hyperparameters learning rate to $10^{-8}$ and regularization weight to $\lambda = 3 \times 10^{-8}$.

### D.2 Alternating optimization - Implementation details

To perform alternating optimization as described in Section 6.1 we run SGD with a learning rate of $5 \times 10^7$ and regularization weight $\lambda = 10^{-4}$ during the reconstruction steps and a learning rate of $5 \times 10^{-11}$ during the motion estimation step. In both steps the loss is the MSE between predicted and given measurement. The optimization process is capped at 500 iterations, but it terminates early if the reconstruction loss falls below the threshold of $e^{13}$.

After alternating optimization we perform L1-minimization from scratch based on the estimated motion parameters. We run 50 steps with SGD and a learning rate of $5 \times 10^7$ and regularization weight $\lambda = 10^{-3}$.

### D.3 E2E stacked U-net - Implementation details

We train the stacked U-net with self-assisted priors proposed in Al-Masni et al. [1] as an example from the class of methods for end-to-end motion correction for 3D MRI. Their method consists of a stack of two U-nets that are trained to map the absolute value of a zero-filled (ZF) reconstruction of the motion corrupted and undersampled measurement to the absolute value of the motion-free image enabling end-to-end motion correction with a single forward pass through the network. The network operates slice-wise however neighboring slices are provided as context at the network input to account for the 3D nature of the problem.

We train the E2E stacked U-net on the same dataset and in a similar way as we trained the U-net on motion-free data in Section D.1.1. In every step one of the 40 3D training volumes is loaded from which the fully sampled target volume is computed as well as three different input volumes - the motion-free undersampled ZF reconstruction and two different motion-corrupted undersampling ZF reconstructions generated with the inter-shot motion simulation described in Section 6.1 with a random level of motion severity and a fixed motion severity of level 1 respectively. For each training example and level of motion severity a fixed set of three different motion trajectories is considered during training.

We include the task of motion-free and motion severity of level 1 reconstruction in every training step to ensure that the model does not loose too much performance when reconstructing motion-free data compared to a model trained only for motion-free reconstruction and to bias the model towards performing well for very mild motion artifacts as it is the regime in which end-to-end methods can still provide reasonably good results, whereas for severe motion the reconstruction results are far from a potential use for medical diagnosis. See Figure 7 for examples of mild and very severe motion.

For each of the three degraded volumes 20 random slices are selected in each plane $r_x \times r_y, r_x \times r_z$ and $r_y \times r_z$ resulting in $3 * 3 * 20 = 180$ pairs of undersampled and potentially motion corrupted training inputs and motion-free fully-sampled training targets per training step. As we train with a batch size of 20, network parameters are updated 9 times per training step and $40 * 9 = 360$ times per epoch.

We train the model for 230 epochs with the L1 loss from equation (37) and the Adam optimizer with a learning rate of $3 \times 10^{-4}$ which is decayed twice by a factor of 10 at epochs 200 and 220 and pick the model from the epoch with the highest reconstruction PSNR for motion-free reconstruction on the validation set.

We adopt the network design from Al-Masni et al. [1], where we set the number of channels in first layer of both U-nets to 64 resulting in a total of 15.9M network parameters. We used instance norm instead of batch norm in our network as we found it to give more stable results across levels of motion severity.

The final motion-corrected reconstructions are obtained by reconstructing the 3D volume slice-wise in the axial $r_x \times r_y$ plane. Before the PSNR scores shown in Figure 4 are computed the reconstruction is aligned with the motion-free reference volume.

### D.4 MRI sequence parameters for real motion experiments

In this section we provide additional information regarding the acquisition of our own data used in Section 6.3. This study was exempt from Institutional Review Board (IRB), but potential risks were disclosed to the subject and experiments were conducted with informed consent. Data was acquired on a Ingenia Elition 3.0T X scanner (Philips Healthcare, Best, The Netherlands) using the standard 16-channel dStream HeadSpine coil array, where $C = 13$ channels were used during acquisition. We perform 3D T1-weighted Ultra-fast Gradient-echo (TFE) imaging with a 1mm isotropic resolution and a matrix-size of $k_x \times k_y \times k_z = 222 \times 236 \times 512$, an undersampling factor of 4.94 and a linear sampling trajectory illustrated in Figure 1 (d). We subsample the data by a factor of two along the fully-sampled frequency encoding dimension $k_z$ to obtain a similar field of view as in our training data with $k_z = 256$. In one shot 204 lines in the k-space are acquired resulting in a total number of 52 shots. See Table 1 for an overview of all sequence parameters.

Table 1: Sequence parameters used in the real motion experiments.

| Parameter | Value |
| --- | --- |
| Sequence | 3D T1-TFE |
| Flip angle (deg) | 8 |
| TR (ms) | 6.7 |
| TE (ms) | 3.0 (shortest) |
| TFE prepulse / delay (ms) | non-selective invert / 1060 ms |
| Min. TI delay (ms) | 707 |
| TFE factor | 204 |
| TFE shots | 52 |
| TFE dur. shot / acq (ms) | 1742 / 1347 |
| Shot interval (ms) | 3000 |
| Sampling | Cartesian |
| Under-sampling factor | 4.94 |
| Half-scan factor Y / Z | 1 / 0.85 |
| Number of auto-calibration lines | 37 |
| Profile order | random |
| FOV (FH x AP x RL, mm) | 256 x 221 x 170 |
| Acquisition matrix | 256 x 221 |
| Fold-over direction | AP |
| Fat shift direction | F |
| Water-fat shift (pixels) | 1.6 |

## E  Additional experimental results

In this section we provide additional ablation studies and further qualitative examples to complement the experimental results presented in Sections 6.1, 6.2, and 6.3 on simulated inter-/intra-shot motion and real-motion respectively.

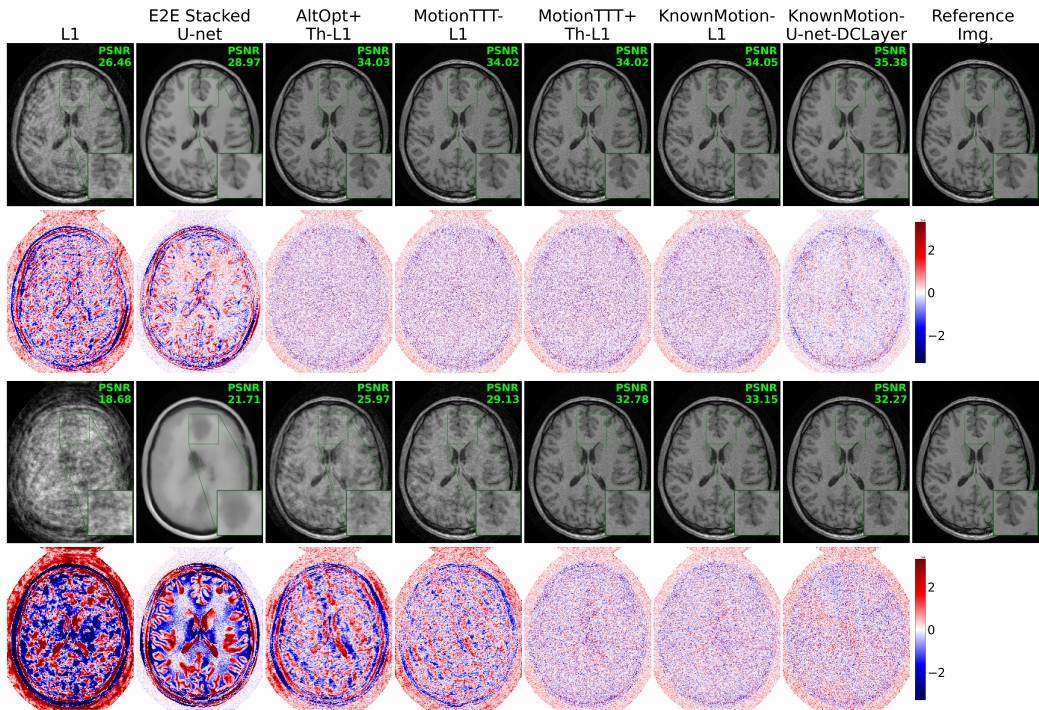

Figure 7: Visual comparison with reconstructions and difference images for simulated motion of severity level 1 (first two rows) and severity level 9 (last two rows) for all methods presented in Figure 4.

## E.1 Additional inter-shot results

We present additional results on inter-shot motion estimation, analysing reconstructions at different levels of motion severity. Qualitative comparisons for mild and severe cases highlight the strengths and limitations of the motion estimation methods used.

### E.1.1 Additional qualitative results

Figure 5 in Section 6.1 in the main body shows reconstructed images for a simulated motion severity level 9.

Figure 7 shows additional reconstruction results at a lower severity level (level 3); for this lower motion severity level both *MotionTTT+Th-L1* and *AltOpt+Th-L1* achieve results comparable to *KnownMotion-L1*.

Figure 8 shows an example of predicted motion and corresponding DC loss for a simulated inter-shot motion scenario at severity level 9. The DC loss effectively detects incorrectly estimated motion states, highlighting their locations. This capability is particularly useful for DC thresholding, which improves robustness, as discussed in Section 6.1. In this scenario, MotionTTT failed in only one shot, whereas the AltOpt method failed in 12 shots, further demonstrating the effectiveness of the MotionTTT approach.

### E.1.2 Comparison of Computational Time

Beyond reconstruction performance, computational time is an important factor for real-world applications. If the reconstruction process is too slow, the algorithm may be impractical for clinical use. The following table summarizes the average computational times for the AltOpt and MotionTTT methods, tested on an Nvidia A100 GPU. As shown in the table, our MotionTTT method is approximately 6 times faster than AltOpt when early stopping is applied. Without early stopping, AltOpt is even 10 times slower than MotionTTT.

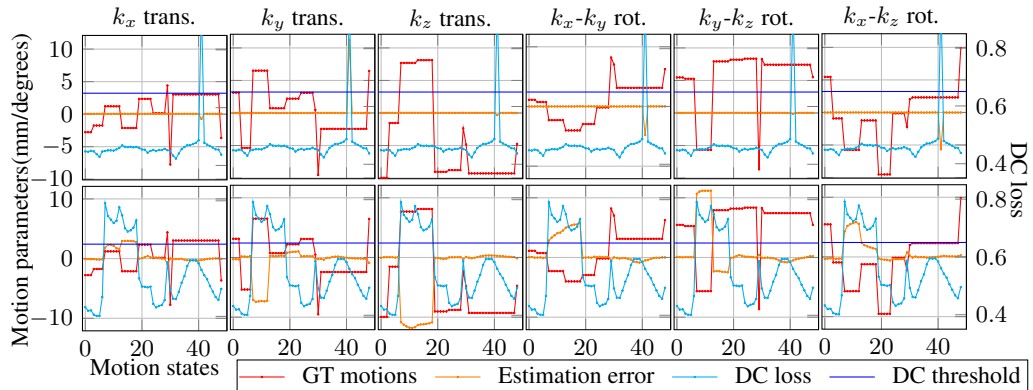

Figure 8: Example of a simulated inter-shot motion trajectory (GT motions) for severity level 9 corresponding to the example in Figure 5. Our MotionTTT (first row) estimation fails only for a single motion state, whereas alternating optimization (second row) fails at recovering several motion states. The corresponding DC losses and the DC threshold indicate which shots are excluded from the reconstruction.

| Method | AltOpt(full run) | AltOpt (early stopping) | MotionTTT |
|---|---|---|---|
| Average Running Time | 4 hours 15 minutes | 2 hours 39 minutes | 25 minutes |

### E.1.3  Ablation studies on the acceleration factor

In this section we investigate the role of the undersampling factor on the ability of MotionTTT to estimate inter-shot motion. We re-train the U-net on two additional Cartesian undersampling masks with acceleration factors $R = 2, 8$ in addition to the existing results with acceleration factor $R = 4$ (see Figure 9 a,b,c for the masks). Figure 10 shows the reconstruction performance in PSNR based on motion parameters estimated by our MotionTTT compared to ground truth motion over three levels of motion severity and the three acceleration factors.

As expected, the overall performance decays with increasing acceleration factors and motion severities. For mild and moderate motion, MotionTTT achieves highly accurate motion estimation for all acceleration factors indicated by the vanishing performance gap relative to using ground truth motion.

For the most severe motion, a small performance gap exists for all acceleration factors due to incorrectly estimated motion states that are discarded from the final reconstruction via DC loss thresholding. In fact, under severe motion an average of 2.5/100, 2.0/50 and 0.12/25 shots have to be discarded for acceleration factors 2,4 and 8.

We conclude that MotionTTT can achive highly accurate motion parameter estimation robustly across different acceleration factors. We attribute the slight increase in discarded motion states for smaller acceleration factors to the increased complexity of the optimization problem as the number of unknown motion states to be estimated increases linearly in the number of acquired shots.

### E.2  Ablation studies for intra-shot motion estimation

In this section we present addition ablation studies for the choice of the number of motion states $N_{\text{splits}}$ estimated per shot and the sampling order used in our experimental results on simulated intra-shot motion estimation in Section 6.2. We also show reconstruction results from the experiments in the main body in Figure 11.

### E.2.1  Number of motion states per shot

We start with an ablation study on the choice of the hyperparameter $N_{\text{splits}}$ that determines the number of motion states that are introduced at the end of phase 1 of MotionTTT's optimization scheme outlined in Section 4 for each shot that exhibits a data consistency loss larger than a certain threshold.

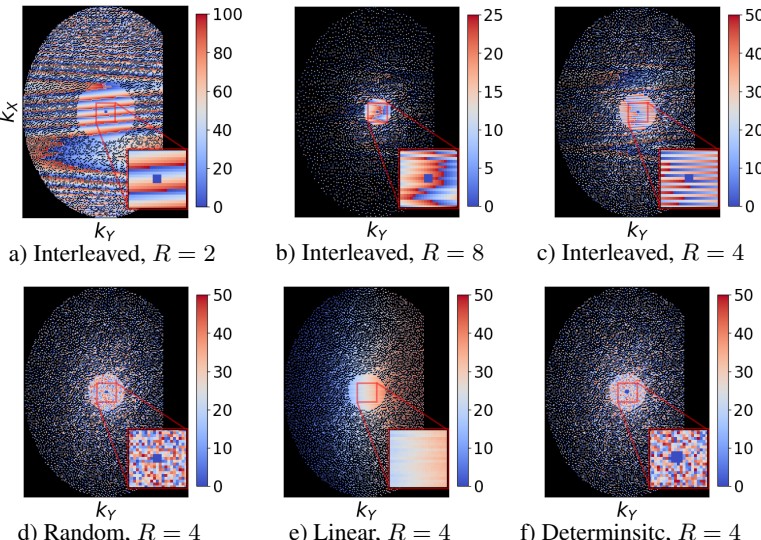

a) Interleaved, $R = 2$  b) Interleaved, $R = 8$  c) Interleaved, $R = 4$

d) Random, $R = 4$  e) Linear, $R = 4$  f) Determinsitc, $R = 4$

Figure 9: Undersampling masks used in the ablation studies in Appendix E for different acceleration factors $R \in \{2, 4, 8\}$ with corresponding number of shots $\{25, 50, 100\}$ such that a constant number of k-space lines is acquired per shot. The color coding illustrates the sampling trajectory (interleaved, random or linear) indicating which k-space lines are sampled within the same shots.

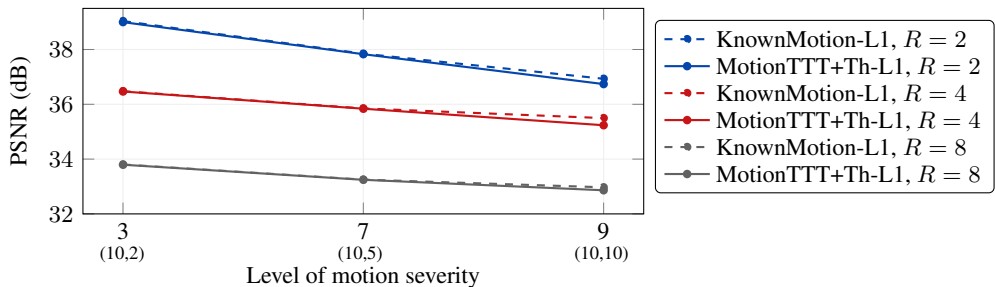

Figure 10: Performance of L1-minimization with known motion versus with motion estimated by MotionTTT over three different levels of motion severity (defined in Figure 4) for acceleration factors $R = 2/4/8$. Results are averaged over 4 validation examples with 2 motion trajectories each.

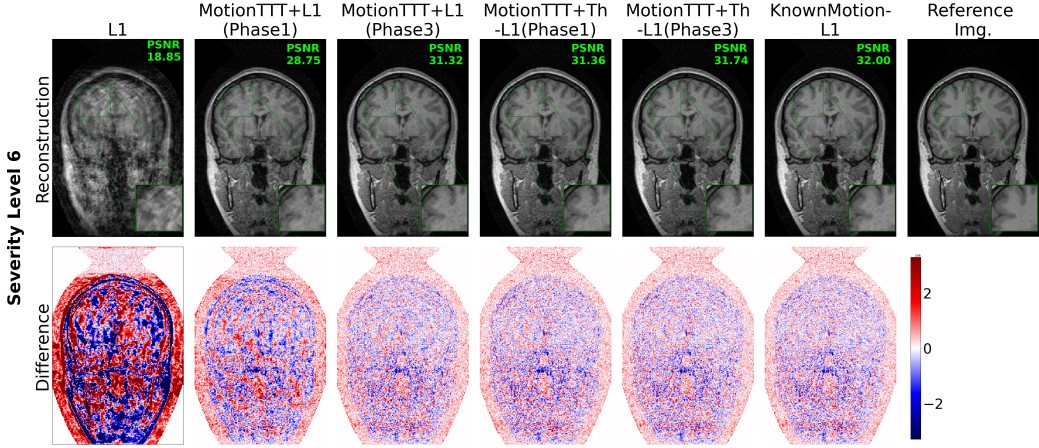

Figure 11: Reconstructions and difference images for simulated *intra-shot motion* of severity level 6.

Table 2: Reconstruction performance of *MotionTTT+Th-L1* in PSNR averaged over four validation examples and five randomly generated motion trajectories per example together with the average number of discarded k-space lines due to DC loss thresholding before the final reconstruction. We compare the performance of four different sampling orders random, deterministic, interleaved and linear for intra-shot motion correction.

| Sampling order | Random | Deterministic | Interleaved | Linear |
|---|---|---|---|---|
| PSNR (dB) | 36.05 | 36.06 | 35.67 | 33.52 |
| # of discarded lines | 197 | 177 | 516 | 3389 |

As discussed in Section 6.2, the choice of this hyperparameter trades off the irreducible discretization error versus the available signal per estimated motion state and the computational complexity.

The discretization error results from estimating only $N_{\text{splits}}$-many motion states for a shot that when affected by intra-shot motion exhibits a distinct motion state for each k-space line (182 in our setup) acquired within this shot. The error decreases with increasing $N_{\text{splits}}$ and depends on severity of motion as fast movements with a large amplitude result into a large discretization error.

On the other hand, increasing $N_{\text{splits}}$ decreases the amount of k-space signal per estimated motion state potentially leading to more incorrectly estimated motion states. Additionally, doubling the number of motion states $N_{\text{splits}}$ doubles the computational complexity of phase 2 if the available hardware does not allow for parallelization (see Appendix D.1.3 for a discussion of computational aspects).

To find the value for $N_{\text{splits}}$ that trades off those two effects, we conduct the following experiment. We simulate motion trajectories containing 5 intra-shot motion events following Section 6.2 with maximal motion $M_{max} = 5$. We assume that all motion parameters are known except the ones during the intra-shot events. We then apply *MotionTTT+Th-L1* with a random sampling order to estimate the motion states during intra-shot motion for different levels of discretization defined by the number of motion states per shot $N_{\text{splits}} \in \{5, 10, 20\}$. This simulates the case where phase 1 of the intra-shot motion estimation scheme perfectly estimates the inter-shot motion parameters and detects all shots corrupted by intra-shot motion for which then motion parameters are estimated during phase 2. This allows us to focus the evaluation solely on the ability of the method to perform intra-shot motion estimation independently from the performance in phase 1.

We obtain PSNRs averaged over the four examples in the validation set and five simulated motion trajectories per example as 35.84, 36.05 and 36.07 for $N_{\text{splits}} = 5, 10$ and 20. Further, the average numbers of k-space lines that are discarded due to DC loss thresholding before the final reconstruction are 289, 197 and 302 out of $5 * 182 = 910$ lines affected by intra-shot motion.

For $N_{\text{splits}} = 5$ we obtain the lowest performance, which we attribute to a large discretization error, which even results into more motion states with a DC loss above the threshold than for $N_{\text{splits}} = 10$. For $N_{\text{splits}} = 10, 20$ we obtain the same performance, however due to different reasons. While for $N_{\text{splits}} = 20$ a smaller discretization error can be achieved, more motion states are estimated incorrectly and are discarded due to a more difficult optimization problem as more motion states are estimated with less signal per motion state. We use $N_{\text{splits}} = 10$ for our experiments in Section 6.2 as it corresponds to smaller computational costs for the same performance.

### E.2.2    Sampling order

Next, we present an ablation study on the role of the sampling order for the ability of MotionTTT to estimate intra-shot motion as in Section 6.2.

In order to focus our evaluation on the performance for intra-shot motion estimation, we consider the same experimental setup as in Appendix E.2.1, where we simulate motion trajectories containing 5 intra-shot events with maximal motion $M_{max} = 5$, apply MotionTTT only to the motion states affected by intra-shot motion and assume all other motion states to be known. We fix the number of motion states per shot to $N_{\text{splits}} = 10$ and report the performance for different sampling orders including the interleaved and random sampling orders used for the inter- and intra-shot motion experiments, as well as a linear and a deterministic order. See Figure 9 masks c)-e) for visualizations.

The linear order acquires k-space lines according to their $k_y$ index from low to high. Hence, all shots towards the start and end of the scan only contain high-frequency components. The interleaved order distributes high-/low-frequency components evenly across shots, but within one shot the order follows the $k_x$ index from low to high. Hence, all k-space lines acquired towards the start and end of one shot are high-frequency components. The random order acquires all k-space lines at a random order. The deterministic order we designed such that the average distance to the center of the k-space of two k-space lines acquired after each other is approximately constant, which ensures that after the acquisition of a high-frequency components a low-frequency component is acquired. The random, interleaved and deterministic orders all start by sampling the $3 \times 3$ center of the k-space.

Table 2 shows the average performance of *MotionTTT+Th-L1* in PSNR for each sampling order as well as the average number of k-space lines that are discarded due to the DC loss thresholding before the final reconstruction.

While random and deterministic orders perform equally, the interleaved order results in more incorrectly estimated motion states. For the linear order we obtain a significant loss in performance, and over a third of the 9078 lines acquired in total are discarded including many lines that are not affected by intra-shot motion and for which we assume knowledge of the true motion state in this experiments. In fact, under the linear order entire shots at the start and end of acquisition are discarded due to their large DC loss despite assuming knowledge of the motion states. We attribute this to the U-net used during MotionTTT reconstructing high-frequency components not as faithfully as low-frequency components, and conclude that in order to reliably estimate the motion state of a batch of measurements the batch needs to contain low-frequency components.

The deterministic order fulfills this requirement by design and the random order with high probability as for the number of motion states per shot $N_{\text{splits}} = 10$ the smallest batch of measurements we consider consists of about 18 k-space lines, which is large enough to contain both high- and low-frequency components especially as the center of the k-space is much more densely sampled than towards the edge. In our experiments we hence use a random sampling scheme due to its ease of implementation on top of any arbitrary design of the undersampling mask.

### E.3 Results from Real Motion Experiments

We now discuss the real motion experiments from Section 6.3 in more detail.

**Visual reconstruction analysis.** Figure 12 shows the reconstruction of the volume in three orthogonal planes (axial, sagittal, and coronal). For mild motion, artifacts are still noticeable, particularly in the $k_y$-$k_z$ plane. However, the *MotionTTT-L1* method effectively mitigates most of these artifacts, resulting in a much clearer reconstruction than the other methods. For strong motion, many details across all three planes are entirely occluded, yet the *MotionTTT-L1* method significantly improves image quality, restoring visibility to numerous details that would otherwise be lost due to severe motion.

**Estimated motion parameters and DC loss.** Figure 13 shows the estimated motion parameters as well as the DC loss (5) per state for the strong motion example. The number of motion states is 69, which results from starting phase 1 of MotionTTT with 52 motion states (one per shot), of which four exceeded the threshold after phase 1. As the number of additional motion states per corrupted shot is $N_{\text{splits}} = 10$, phase 2 and 3 continue with 92 motion states from which 23 were discarded during the final DC loss thresholding resulting in the 69 motion states depicted in Figure 13. The areas shaded in gray correspond to the motion states that were added after phase 1, i.e., intra-shot motion states, whereas the white areas correspond to motion states each pertaining a single shot. Consequently, the predicted motion and the DC loss of MotionTTT (phase 1) is constant within the gray areas as intra-shot motion estimation is performed starting from phase 2. As we can see intra-shot estimation, i.e., MotionTTT (phase 3) significantly reduced the DC loss compared to phase 1 and the estimated motion is reasonably smooth, indicating the benefit of estimating intra-shot motion in practice.

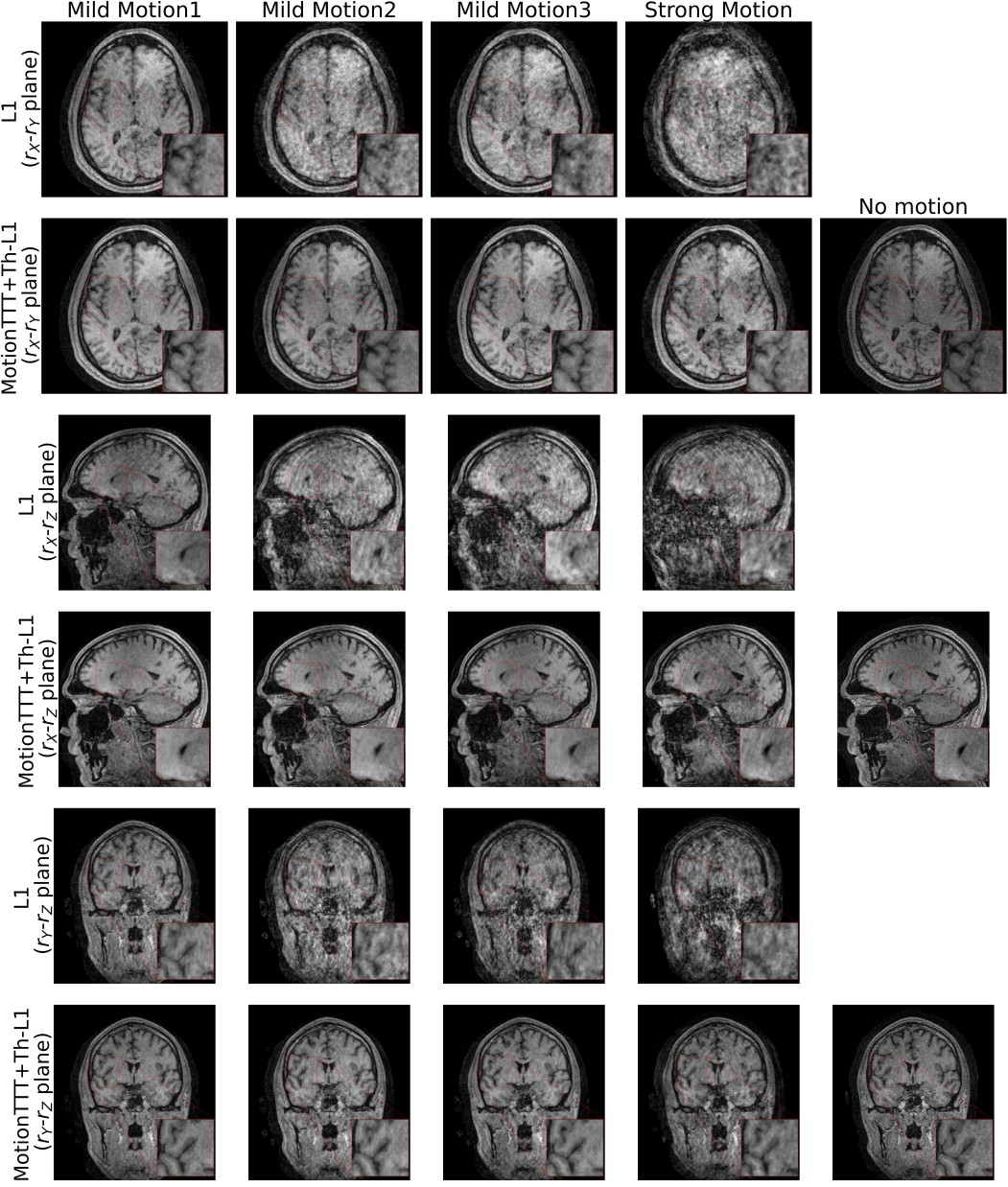

Figure 12: Visual comparison in three planes of reconstructed data corrupted with real mild/strong motion and no motion. Our *MotionTTT-L1* achieves significant gains in image quality compared to *L1* without any motion correction.

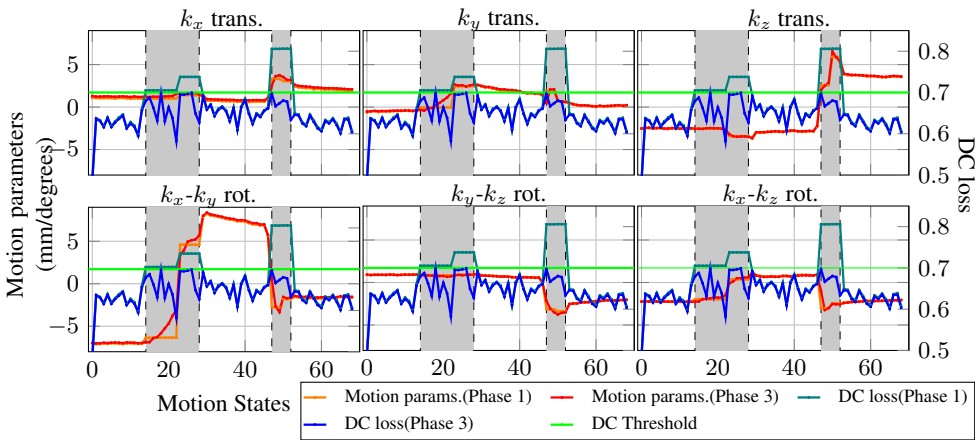

Figure 13: Estimated motion parameters and DC loss of MotionTTT phase 1 and 3 for the data from Figure 12 (strong motion). Gray areas indicate intra-shot motion states and white areas motion states that correspond to a single shot each.

