# OpenReview forum: "MotionTTT: 2D Test-Time-Training Motion Estimation for 3D Motion Corrected MRI"
_NeurIPS.cc/2024/Conference — NeurIPS 2024 poster_

### Official Review · Reviewer_pjnK · 2024-07-10

**Soundness:** 3
**Presentation:** 3
**Contribution:** 3
**Rating:** 6
**Confidence:** 4

**Summary:**

This work proposes a novel deep learning-based framework for recovering high-quality 3D MR images from undersampled and motion-corrupted k-data. The proposed approach is well motivated and technically sound. The authors perform extensive experiments on simulated and real MR datasets, which confirm the effectiveness of the proposed framework.

**Strengths:**

**Motivation and significance**

Motion correction is an important issue in the field of MR imaging. The proposed method removes the reliance on motion simulation by training a neural network that reconstructs high-quality MR image from under-sampled k-data in advance. This new paradigm significantly improves the robustness of the reconstructed images.

**Technical solid**

In the steps 2 of test-time training for motion estimation, the MRI acquisition knowledge, such as forward model, sampling trajectory, are effectively integrated into the framework, which improve the reliability of the reconstruction.

**Clarity and organization**

This submission is well-written and easy to follow.

**Experimental evaluation**

The authors perform experimental evaluations on simulation and real-world datasets. I am pleased to see the experiments based on the real-world dataset. I think it greatly improves this paper.

**Weaknesses:**

For this submission, I have a few minor suggestions as follows.

**Various types of rigid motion**

In line 205, the authors show that random rigid motions ($M_\text{max}=[2,5,10]$) are simulated. However, the rigid motion in the real world could follow some patterns, such as involuntary motion and abrupt motion. I think it is better to test these different movements.

**Compared methods**

For supervised methods, only U-net is used as a baseline. Advanced supervised models [1][2] are not discussed. Furthermore, motion correction methods [3] based on diffusion models are not compared.

> [1] Han Y, Yoo J, Kim H H, et al. Deep learning with domain adaptation for accelerated projection‐reconstruction MR[J]. Magnetic resonance in medicine, 2018, 80(3): 1189-1205.

> [2] Liu J, Kocak M, Supanich M, et al. Motion artifacts reduction in brain MRI by means of a deep residual network with densely connected multi-resolution blocks (DRN-DCMB)[J]. Magnetic resonance imaging, 2020, 71: 69-79.

> [3] B. Levac, A. Jalal, and J. I. Tamir. “Accelerated Motion Correction for MRI Using Score-Based 388 Generative Models”. ISBI 2023.

**Some typos**

For example, the symbol $\mathcal{E}$ in Eq. 3 is not defined.

**Questions:**

See the section of Strengths and Weaknesses, please.

**Limitations:**

The authors mention enough limitations of their work.

---

> ### Author Rebuttal · Authors · 2024-08-07
>
> We thank the reviewer for the feedback and the positive evaluation of our work. In the following we address the weaknesses in the order as pointed out by the reviewer.
> - **Weakness 1, various types of rigid motion:** We agree that in practice motion can be categorized into different types. However, the range of different motions is large varying with patient population and condition. Also we are not aware of good available models of the different types of motion based on real measurements in the literature. Hence, it is common to evaluate with randomly simulated motion (see line 202 for examples).
> Note that for the experiments in Figure 3 we simulated a total of 100 different motion trajectories with up to 10 randomly generated motion events per trajectory covering a wide range of possible motions.
> In general, we expect our method to work well for different types of motion as different to previous deep learning based approaches it was not trained on a particular type of simulated motion.
> - **Weakness 2, compared methods:** [1] proposes a domain adaptation method for reconstructing radial MRI in a limited data regime, but does not discuss motion correction. [2] proposes a novel network architecture for end-to-end motion artifact reduction but only in 2D and in general it has been found that end-to-end methods, while being significantly faster, result in inferior reconstruction quality compared to reconstruction based on previously estimated motion parameters like ours. See e.g. Fig. 3 in [4] and Fig. 7 in [5].
> While diffusion model based motion correction is an interesting direction, the reason why we do not compare to [3] is that the proposed method can only operate 2D motion, while we investigate motion estimation in 3D. Directly extending the method to 3D requires training a diffusion model on entire 3D volumes, which is computational difficult and would require much larger 3D datasets. At the same time it is unclear how to use a 2D diffusion model to estimate motion in 3D.
> - **Weakness 3, some typos:** Thanks, we fixed it.
>
> [4] Haskell et al. “Network Accelerated Motion Estimation and Reduction (NAMER): Convolutional Neural Network Guided Retrospective Motion Correction Using a Separable Motion Model”. In: Magnetic Resonance in Medicine (2019).
> [5] Hossbach et al. “Deep Learning-Based Motion Quantification from k-Space for Fast Model-Based Magnetic Resonance Imaging Motion Correction”. In: Medical Physics (2023).

---

> > ### Comment · Reviewer_pjnK · 2024-08-13
> >
> > Thank you for the detailed response. I have also reviewed the comments from other reviewers. While I believe this paper makes a significant contribution to 3D motion correction in MRI, there are some limitations, such as the limited comparison of methods. Therefore, I recommend a weak acceptance for this submission.

---

> > > ### Author Response · Authors · 2024-08-14
> > >
> > > Thanks for noting that our paper makes a significant contribution to 3D motion correction, and many thanks for the reply.
> > >
> > > Regarding comparing to more methods, we would have been happy to compare to more methods, but found no suitable 3D methods other than the one we compare to (alternating minimization). Please not that all methods you mentioned are for 2D motion correction, and they do not extend to 3D in a straightforward manner. We think it can’t be expected to extend other work significantly in order to generate new baselines, this would be independent contributions. Motion estimation and correction in 3D is extremely difficult, this is testified by the fact that the vast majority or papers only studies 2D motion reconstruction, even though motion in MRI always occurs in 3D space. Please let us know if you have any baseline in mind that has been applied to 3D motion estimation or correction by the papers that proposed the method.

---

### Official Review · Reviewer_H96H · 2024-07-12

**Soundness:** 2
**Presentation:** 1
**Contribution:** 2
**Rating:** 3
**Confidence:** 3

**Summary:**

This paper proposes a motion correction MRI reconstruction algorithm for 3D brain MRI. The proposed technique consists in a deep learning-based estimation of rigid motion parameters, which allows to correct the k-space before a final reconstruction.
Estimation of motion parameters are based on a single optimization step freezing the reconstruction UNet parameters, and assuming loss will only (or mainly) be influenced by motion.

**Strengths:**

The paper addresses an important problem, namely motion correction, using a fully data-driven approach. Moreover, the originality of the approach lies in tackling this problem directly on the 3D frequency (or k-)space , when most deep-learning techniques are focusing on 2D acquisitions.

**Weaknesses:**

The overall structure of the paper is quite confusing, which makes it hard to follow. Experiments and alternative techniques seem to be presented throughout the result section.
A proper ablation study seems required, in order to assess the added-value of each step. Proper baseline also seems necessary especially regarding the reconstruction step or the removal of motion artifacted lines, as more recent techniques could have been used as SOTA.
The overall motion estimation relies on a first 2D reconstruction pipeline however the performance of the technique seems to be lower than a standard L1 reconstruction technique. If so, how could the authors be sure that motion parameters are not biased or affected by the low performance of the 2D reconstruction pipeline.
Why did the authors not try to directly reconstruct from the 3D k-space domain, in order to estimate motion parameters in 3D? Moreover, I believe the authors should also provide further details on they are switching back and forth from 3D k-space domains to 2D, as a simple slicing in a given direction of the 3D k-space is not appropriate.

**Questions:**

Could the authors find another database of 3D data in order to train a 3D reconstruction UNet directly?
Given the proposed technique aims also at suppressing corrupted k-spaces lines, would it be sensible to train a reconstruction technique with higher a acceleration factor?
Would the proposed technique be applicable for other sampling strategies, especially would it be possible to apply on stack of stars acquisitions or other 3D sampling strategy.
Why did the authors not apply other reconstruction (standard L1) reconstruction for the real dataset for comparison? Especially since it was outperforming the UNet approach on simulated motion data.

**Limitations:**

The authors should discuss the risks of hallucinations and bias of the proposed technique.

---

> ### Author Rebuttal · Authors · 2024-08-07
>
> Thanks for the feedback. In the following we address the concerns and questions in the order as pointed out by the reviewer.
> - **Weakness 1, experiments and alternative techniques seem to be presented throughout the result section:** Thanks for the feedback. We will make sure to describe all methods that we compare in the main Figure 3 at the end of the setup Section 5.1 before the results section.
> - **Weakness 2, a proper ablation study seems required, in order to assess the added-value of each step:** In fact, Figure 3 already shows the performances during all steps of MotionTTT. L1 indicates the performance without any motion correction (before applying MotionTTT), MotionTTT-L1 the performance after motion estimation and MotionTTT-L1+Th after motion estimation plus data consistency loss thresholding.
> - **Weakness 3, baselines regarding the reconstruction step or the removal of motion artifacted lines:** Regarding the reconstruction step, our method is agnostic to the method used for reconstruction based on the estimated motion parameters. Hence we do not expect any new insights from trying additional reconstruction methods as a better reconstruction method would lead to a better performance for any motion estimation method.
> &nbsp;
> Regarding baselines for the removal of motion artifacted lines, we are not sure to which method the reviewer would like to see a comparison. Our method estimates motion parameters with a high accuracy and hence will outperform any method that only detects and then removes motion corrupted lines. Only in the presence of severe motion we remove lines that exhibit a large data consistency loss indicating inaccurate estimation of motion parameters, a feature that comes for free with the proposed method. However, also in the most severe cases typically not more than 5% have to be discarded.
> - **Weakness 4, effect of the low reconstruction performance of the U-net on motion estimation:** It is not clear if a better reconstruction performance of the U-net due to an increased amount of training data would further improve the ability to estimate motion parameters especially as our experimental results show, the current reconstruction performance of the U-net already achieves highly accurate motion parameter estimation across a wide range of simulated motion and significantly improved image quality in case of prospectively acquired real motion-corrupted data. Nevertheless, we will investigate if even further improvements can be achieved by training on additional data sources like e.g. 2D brain data.
> - **Weakness 5, why did the authors not try to directly reconstruct from the 3D k-space domain, in order to estimate motion parameters in 3D:** Training 3D models on entire 3D volumes is infeasible in terms of memory and compute requirements. However, we agree that instead of chunking the volume slice-by-slice as we do, one could also train a 3D model on 3D chunks and perform chunk-by-chunk reconstruction.
> We decided to build on the well-established 2D convolutional U-net as it enabled highly accurate motion estimation combined with fast reconstruction and a low network parameter count, which is important as the entire volume has to be reconstructed in every iteration. In the future, this also enables enlarging the training set with 2D data in order to overcome the limited availability of 3D dataset.
> - **Weakness 6, regarding switching between 2D and 3D:** As the reviewer pointed out correctly, slicing the k-space in arbitrary directions is not appropriate. Hence, we only slice the zero-filled reconstruction in the image domain at the network input and add slices together at the network output before applying the 3D fourier transform to obtain the reconstructed k-space data. We will emphasize this more clearly in the Section 4, step 2: Test-time-training for motion estimation.
> - **Question 1, existence of another database of 3D data to train a 3D reconstruction UNet directly:** To the best of our knowledge the used dataset is the largest 3D brain dataset containing the original k-space measurements publicly available. While other smaller ones exist, we do not believe that even if compute would not be a problem the amount of training examples would suffice to train a 3D model on entire volumes.
> - **Question 2, training a reconstruction technique with higher acceleration factor to compensate for thresholding:** We only threshold a relatively small number of lines, typically not more than 5% of the acquired lines even under the most severe motion. Nevertheless, there is a discrepancy between the undersampling mask the U-net was trained on in the absence of motion and the undersampling of the motion-corrected network inputs due to rotations in the k-space as discussed in the paper in Section 5.2. However, both changes in the mask due to thresholding and rotations are motion specific and hence difficult to train on without simulating motion.
> Hence, in the paper we use L1-minimization for reconstruction, which is mostly independent of changes in the mask.
> - **Question 3, application to other 3D sampling strategies (stack of stars):** See author rebuttal point 1.
> - **Question 4, missing L1 reconstruction for the real-motion data:** We do show the results for MotionTTT+Th-L1 in Figure 6.
> - **Limitation, risks of hallucinations and bias:** As currently the final reconstruction results of our method are based on L1-minimization no learning based hallucinations can occur. If the motion parameter estimation is off, the result will be a reconstruction with motion artifacts similar to when no motion correction is applied. We are not sure what type of bias the reviewer is referring to.
>
>
> We hope those responses address the reviewer's concerns and the reviewer considered raising their score.

---

### Official Review · Reviewer_4RZt · 2024-07-14

**Soundness:** 2
**Presentation:** 2
**Contribution:** 2
**Rating:** 4
**Confidence:** 2

**Summary:**

This paper proposes a method for estimating the motion of patients, such that accurate motion-corrected images could be reconstructed. The key idea is that a neural network trained for motion-free reconstruction has a small loss if there is no motion, thus optimizing over motion parameters passed through the reconstruction network enables estimation of the motion.

**Strengths:**

- The estimation of motion of patients is important in MRI measurement. This paper is a good trial to address this problem.

**Weaknesses:**

- There is no technical novelty in this work. It introduces some existing modules to address the proposed problem.
- The experiments are weak to demonstrate the efficacy of the proposed method.
  1）There lacks strong baselines and there are no compared methods. Although according to the authors, there are no other methods designed for 3D rigid motion estimation for 3D motion-corrected MRI, there should be some methods that could reconstruct the images.
  2) The data are little for evaluating the methods, and more averaged quantitative results are suggested to report.

**Questions:**

See the weaknesses.

**Limitations:**

Yes, limitations are discussed.

---

> ### Author Rebuttal · Authors · 2024-08-07
>
> We thank the reviewer for the feedback. In the following we address the concerns in the order as pointed out by the reviewer.
> - **Weakness 1, introduction of existing modules and lack of novelty:** We would like to point out that combining a 2D reconstruction network trained on motion-free data with test-time-training for motion estimation in 3D has not been proposed before, neither a closely related method. Our work is the first that enables efficient motion estimation in 3D based on a 2D network trained on motion-free measurements. As discussed in our related work section, previous work on deep learning based motion estimation has exclusively focused on the 2D case and methods that are required to simulate motion during training and hence are specific to the type of motion used during training. Thus the work contains significant technical novelty.
> - **Weakness 2 part 1, lack of further baselines in particular motion corrupted image reconstruction networks:** While we are not aware of any deep learning based baselines for 3D motion estimation, there are indeed methods for 3D end-to-end reconstruction from motion corrupted volumes as discussed in our related work section, second paragraph. As we note there, this class of methods is well known to perform poorly relative to methods that perform motion estimation and then reconstruct, like our method. See Fig. 3 in [1] and Fig. 7 in [2].
> - **Weakness 2 part 2, average quantitative results over more test examples:** For a given method each result in Fig. 3 in the paper is averaged over 5 test examples with each 2 independent motion trajectories. As we consider 10 different levels of motion, we evaluate in total over 100 different motion trajectories each consisting of up to 10 randomly generated motion events. Regarding the required number of test examples, for image reconstructions tasks this number is typically lower than in other ML domains like computer vision as one test image contains already many different structures and details that the method needs to recover in order to achieve a high image quality score. Especially, in 3D the amount of data per test volume is large and comparatively small test sets are often used (e.g. 7 volumes used in [3]).
>
> We hope this addresses the reviewer's concerns and if yes, that the reviewer considers raising their score.
>
>
> [1] Haskell et al. “Network Accelerated Motion Estimation and Reduction (NAMER): Convolutional Neural Network Guided Retrospective Motion Correction Using a Separable Motion Model”. In: Magnetic Resonance in Medicine (2019).
> [2] Hossbach et al. “Deep Learning-Based Motion Quantification from k-Space for Fast Model-Based Magnetic Resonance Imaging Motion Correction”. In: Medical Physics (2023).
> [3] Johnson and Drangova. “Conditional Generative Adversarial Network for 3D Rigid-Body Motion Correction in MRI”. In: Magnetic Resonance in Medicine (2019).

---

> > ### Comment · Reviewer_4RZt · 2024-08-14
> >
> > Thanks to the authors for the rebuttals. Some of the concerns are addressed but some remain. After reading the comments of all reviewers, I keep my original rating.

---

> > > ### Author Response · Authors · 2024-08-14
> > >
> > > Can you be specific which concerns remain? You point out that “this paper is a good trial to address this problem” but at the same time argue that the work lacks novelty and that there “should be other methods”, without pointing to any specific work in the literature. If you have any concrete concerns that remain, we would be happy to clarify.

---

### Official Review · Reviewer_fV1o · 2024-07-14

**Soundness:** 2
**Presentation:** 3
**Contribution:** 3
**Rating:** 4
**Confidence:** 4

**Summary:**

The paper presents MotionTTT, a deep learning-based method for estimating and correcting rigid motion in 3D MRI images. The approach leverages a neural network pre-trained for 2D motion-free image reconstruction and employs test-time-training (TTT) to estimate motion parameters from motion-corrupted 3D measurements. The effectiveness of the method is demonstrated through evaluations on both simulated and real datasets.

**Strengths:**

The paper proposed a novel approach for 3D rigid motion estimation using a neural network trained on 2D motion-free images and TTT for motion parameter estimation. The application of test-time-training for motion estimation is a novel contribution that effectively addresses motion artifacts in 3D MRI images, and outperformed classical alternating optimization methods in terms of speed and accuracy, especially under severe motion conditions.

**Weaknesses:**

The theoretical foundations of MotionTTT are generally robust, but there are notable gaps. The method builds on existing neural networks for 2D image reconstruction, extending them to handle 3D rigid motion estimation.

During the pre-training step, $f_\theta$ is trained to map under-sampled k-space data back to fully sampled k-space data. The authors show that under-sampled k-space data with motion (where certain regions of the k-space undergo rotation and phase shifting) results in higher reconstruction loss for $f_\theta$. However, this indirect optimization lacks theoretical proof and needs more robust experimental validation.

For instance, the performance of this method may heavily depend on the under-sampling pattern of the k-space (such as the undersampling factor and the linear sampling trajectory) and the specific regions in k-space that are affected by motion. However, in the experiments, the setup of the undersampling of the k-space was fixed.

A more thorough theoretical and experimental investigation into why this optimization works and under what conditions it is most effective would strengthen the soundness of the method.

**Questions:**

1. The idea behind minimizing loss (4) is not intuitive and clear. How do you prove the effectiveness of using loss (4) to recover and remove the motion corruption? Will the effectiveness be affected by the undersampling factor and trajectory? What if you have a higher or lower factor? or maybe a spiral trajectory? Can you find the optimal factor for your method?
2. Based on the assumption that rigid motion leads to rotation and phase shift in k-space, is the linear trajectory the optimal undersampling trajectory?
3. Do you really need the fully sampled k-space data to learn $f_\theta$?

**Limitations:**

The indirect TTT optimization method lacks theoretical justification. The authors should investigate both theoretically and experimentally why this approach works and under what conditions it is most effective.

The performance of MotionTTT may be heavily dependent on the under-sampling pattern of the k-space and the specific regions affected by motion. The method’s effectiveness might vary significantly with different sampling patterns and motion artifacts.

The method relies on a pre-trained network using fully-sampled data, which may not always be available in practice.

---

> ### Author Rebuttal · Authors · 2024-08-07
>
> Thanks for the feedback and for acknowledging the novelty of our work. In the following we address the weaknesses (W), questions (Q) and limitations in the order as raised by the reviewer.
> - **W 1, lack of theoretical investigation of why the optimization works:**
> To get an understanding on why the proposed optimization works, we developed the following theory for a (very) simplified setup that models our approach. This setup also helps to understand the necessity of defining a learning rate schedule that in the presence of motion explores the loss landscape with an initially large learning rate before gradually decaying it. We will include this result and discussion in the revised version of our paper.
> &nbsp;
> We consider the signal $\mathbf{x} \in \mathbb{R}^n$ that lies in a $d$ dimensional subspace, i.e., the signal is generated as $\mathbf{x} = \mathbf{U} \mathbf{c}$ with orthonormal $\mathbf{U} \in \mathbb{R}^{n \times d}$ and Gaussian vector $\mathbf{c}$.
> &nbsp;
> Let $\mathbf{F}_\mathcal{T}$ be the Fourier matrix with rows chosen in the set $\mathcal{T}$.
> We assume a measurement model, where the signal $\mathbf{x}$ takes on $N_s$ motion states defined by the unknown translations $t _1^\ast,\ldots,t _ {N _ s}^\ast$, and for each translated version of the signal, a set of measurements is collected according to
> $$\mathbf{y _ s} = \mathbf{D} _ {t _ s^\ast, \mathcal{T} _ s} \mathbf{F} _ {\mathcal{T} _ s} \mathbf{x}, \tag{1}$$
> where $\mathbf{D} _ {t _ s^\ast, \mathcal{T} _ s} $ is a diagonal matrix with $e^{i 2 \pi t_s^\ast l / n}, l\in \mathcal{T} _ s$ on its diagonal.
> Here, we use that a circular shift by $t_s^\ast$ in the spatial domain is a multiplication with a complex exponential in the frequency domain.
> We write the acquisition of the entire measurements $\mathbf{y} \in \mathbb{R}^k$ from all motion states as $ \mathbf{y} = \mathbf{D} _ \mathbf{t^\ast} \mathbf{F} _  \mathcal{T} \mathbf{x}.$
> For simplicity we assume the fully sampled case, i.e., $k=n$.
> &nbsp;
> Next, we define a network $f( \mathbf{x}^\dagger) = \mathbf{U} \mathbf{U}^T \mathbf{x}^\dagger$, for which it is straightforward to see that if all motion parameters $\mathbf{t}^\ast$ are known we have that
> $$f( (\mathbf{D} _ {\mathbf{t}^\ast} \mathbf{F})^\dagger \mathbf{y} ) = \mathbf{x}, \tag{2}$$
> i.e., the network reconstructs the signal perfectly.
> The loss function used in the paper for this setting is
> $$\mathcal{L} _ {\text{TTT}} (\mathbf{t}) = || \mathbf{D} _ {\mathbf{t}} \mathbf{F} _ {\mathcal{T}} f( (\mathbf{D} _ {\mathbf{t}} \mathbf{F} _ \mathcal{T})^\dagger \mathbf{y} ) - \mathbf{y} || _ 2^2, \tag{3}$$
> on which we can perform test-time-training with gradient descent with respect to the motion parameters $\mathbf{t}$.
> &nbsp;
> Assuming that $\mathbf{U}$ is a random subspace and that $\mathbf{c}$ is drawn from a Gaussian with identity covariance matrix, it can be shown that the objective function concentrates around
> $$\mathcal{\tilde L} (\mathbf{t}) =  || \mathbf{D} _ \mathbf{t} - \mathbf{D} _ {\mathbf{t}^\ast} \tag{4} ||^2_F.$$
> This expected objective function has a unique minimizer at $\mathbf{t} = \mathbf{t}^\ast$ and is convex in a small region around $\mathbf{t}^\ast$, but not globally.
> &nbsp;
> For setting the number of motion states $N_s=1$ we obtain a simple one-dimensional optimization problem and can inspect the behavior of the loss function graphically. Without loss of generality we set $t^\ast = 0$ and plot the loss $\mathcal{L} _ {\text{TTT}} (t)$ in PDF Fig. 4. As we can see the loss exhibits a global minimum at $t=t^\ast$, but also local minima for $t \neq t^\ast$.
> &nbsp;
> In order to minimize such a function in practice, we defining a learning rate schedule (see Appendix B.2 in the paper), where from the start we explore the loss landscape with a large learning rate if the initial loss is large as this indicates the presence of strong motion and initializing all motion parameters with zero might constitute to a large distance to the true motion parameters. Then, the learning rate is reduced gradually. Also once estimated motion parameters approach the true parameters, we did not observe significant deviations from this solution indicating the existence of a global minimum (or a good local one) in practice.
> - **W 2, experimental investigation under what conditions the method is most effective in terms of undersampling factor and sampling trajectory:** See author rebuttal.
> - **Q 1, why and under what conditions does the method work:** See Weakness 1 and 2.
> - **Q 2, is the linear trajectory the optimal undersampling trajectory:** See author rebuttal. In short, a linear trajectory is not the best choice for motion estimation with our method, an interleaved or random order is better.
> - **Q 3, do you need the fully sampled k-space data to learn $f_\theta$:** We do not, as discussed in Section 6, our method can leverage recent advances in self-supervised training of MRI reconstruction networks. This is a strength of our method, other proposed deep learning based method for (2D) motion correction in MRI rely on simulated motion artifacts during training which require fully sampled k-space data.
> - **Limitations, the method’s effectiveness might vary significantly with different sampling patterns and motion artifacts:** Regarding the role of the sampling pattern see author rebuttal. Regarding the role of motion artifacts we evaluated our method on motion events occurring at random time points with different amplitudes and frequencies resulting in different motion artifacts achieving good performance for all of them. In general, we expect our method to work well for different types of motion as it was not trained on a particular type of simulated motion. Remaining limitations have been discussed above.
>
> We hope the provided theory as well as the additional simulations and clarifications address the concerns of the reviewer and if yes, that the reviewer considers raising their score.

---

### Author Rebuttal · Authors · 2024-08-07

Thanks for the reviews!
We would like to start by emphasizing that our work is the first that enables efficient motion estimation in 3D based on a 2D neural network trained on motion-free measurements. We show that our method can reliably predict motion and that this yields significant improvements in image quality. We provide extensive numerical results, including for simulated motion, as well based on real motion in an MRI scan that we specifically collected for this project. Most existing works for motion estimation and correction are for 2D and do not contain results on real motion in an MRI scanner.

In the following, we address two concerns raised by reviewers fV1o and H96H regarding the role of the sampling trajectory and undersampling factor for the success of our proposed motion estimation method. To address those concerns, we provide additional experimental results in the attached single-page PDF.

**1. The sampling order is important as it is difficult to estimate motion parameters for a batch of k-space measurements that contain only high-frequency components.**

Reviewer fV1o and H96H asked if the effectiveness of MotionTTT is affected by the sampling trajectory (e.g. Cartesian vs. spiral or stack of stars).
Up to trajectory specific additional sources of artifacts like B0 field inhomogeneities that needed to be modeled, the problem of motion estimation does not change with the sampling trajectory, and as our forward model is implemented via the non-linear FFT, our method can process any k-space geometry.

However, the sampling order is important, as we show with an additional experiment. We investigate three different sampling orders for a Cartesian trajectory:  We fix the undersampling mask and change the order at which k-space lines are acquired between interleaved, linear (both have been used already in the paper) and random (see PDF, Fig. 3 c,d,e for a visualization of sampling orders).

For motion severity level 5 we obtain the average (8 instances) reconstruction performance in PSNR for reconstruction based on motion parameters estimated by our MotionTTT vs. ground truth motion as 35.98vs.36.00, 36.27vs.36.28 and 33.16vs.36.99 for interleaved, random and linear sampling orders.
Interleaved and random orders achieve perfect motion estimation, while the linear order leads to a significant gap of almost 3dB.

For a given test example PDF Fig. 1 a,b shows the data consistency (DC) loss of MotionTTT at the first and final iteration.
For the random and interleaved order motion states are estimated accurately resulting in a final DC loss well below our defined DC loss threshold. For the linear order first and last motion states, pertaining to shots that contain only high-frequency, i.e., low-energy components, maintain a high final DC loss and the corresponding estimated motion parameters are off as exemplified with the estimated translation parameter in $k_y$ direction in PDF Fig. 1 c.
We attribute this finding to the U-net used during MotionTTT reconstructing high-frequency components not as faithfully as low-frequency components.

Finally, we would like to note that the sampling order of a Cartesian trajectory can be customized without affecting any other sequence parameters like the shape of the undersampling mask, and hence choosing a, e.g., random order does not limit the generality of our method in the context of Cartesian sampling, which is still most commonly used in practical 3D MRI.

**2. The acceleration factor influences the performance of MotionTTT as a smaller factor leads to overall higher reconstruction performance but also to a more difficult optimization problem as more shots are acquired translating to more unknown motion states to be estimated.**

Reviewer fV1o requested a more robust experimental validation of the conditions in terms of sampling trajectory (see above) and undersampling factor under which MotionTTT works.

To this end, we conducted the following experiment, where we re-train the U-net on two additional Cartesian undersampling masks with acceleration factors R=2,8 in addition to the existing results with R=4 (see PDF Fig. 3 a,b,c for the masks). PDF Fig. 2 shows the reconstruction performance in PSNR based on motion parameters estimated by our MotionTTT compared to ground truth motion over three levels of motion severity and the three acceleration factors.

As expected, the overall performance decays with increasing acceleration factors and motion severities.
For mild and moderate motion, MotionTTT achieves highly accurate motion estimation for all acceleration factors indicated by the vanishing performance gap to using ground truth motion. For strong motion, the best performance is still achieved for the lowest acceleration factor, but an increasing performance gap exists for decreasing acceleration factors due to incorrectly estimated motion states that are then discarded from the final reconstruction via DC loss thresholding. In fact, under severe motion an average of 20.7/100, 2.6/50 and 0.6/25 shots have to be discarded for acceleration factors 2,4 and 8.
We attribute the increasing numbers of incorrectly estimated motion states to the increasing complexity of the optimization problem as the number of unknown motion states to be estimated increases linearly in the number of acquired shots.

We will include those results in the revised version of our paper. We hope that you find the additional experiments helpful and are happy to discuss further.

---

### Author Response · Authors · 2024-08-12
**Discussion period is coming to an end**

We would like to thank the reviewers once again for their insightful comments.
As the open discussion period is coming to an end, we would very much appreciate if the reviewers could kindly respond to our rebuttal, this would provide us with the opportunity to address or clarify any remaining concerns.

---

### Decision · Program_Chairs · 2024-09-25

**Decision:**

Accept (poster)

**Comment:**

The manuscript is on the verge of rejection, yet the authors have presented a compelling rebuttal. This AC acknowledges the complexity of the problem and appreciate this first work towards 3D motion-compensated MRI reconstruction. This is why the AC believes this could be an interesting paper to be discussed at the conference. While this AC is inclined to support weak acceptance, the AC also understands it may not meet the priority threshold given the overall acceptance criteria/ranking. If the paper is accepted, it is with the expectation that the key points from the rebuttal will be effectively integrated into the manuscript to enhance its overall impact.